# LEARNING AN EFFICIENT-AND-RIGOROUS NEURAL MULTIGRID SOLVER

## ABSTRACT

Partial Differential Equations (PDEs) and their efficient numerical solutions are of fundamental significance to science and engineering involving heavy computation. To date, the historical reliance on legacy generic numerical solvers has circumscribed possible integration of big data knowledge and exhibits sub-optimal efficiency for certain PDE formulations. In contrast, AI-inspired neural methods have the potential to learn such knowledge from big data and endow numerical solvers with compact structures and high efficiency, but still with unconquered challenges including, a lack of sound mathematical backbone, no guarantee of correctness or convergence, and low accuracy, thus unable to handle complex, unseen scenarios. This paper articulates a mathematically rigorous neural PDE solver by integrating iterative solvers and the Multigrid Method with Convolutional Neural Networks (CNNs). Our novel **UGrid** neural solver, built upon the principled integration of **U**-Net and Multi**Grid**, manifests a mathematically rigorous proof of both convergence and correctness, and showcases high numerical accuracy and strong generalization power to complicated cases not observed during the training phase. In addition, we devise a new residual loss metric, which enables unsupervised training and affords more stability and a larger solution space over the legacy losses. We conduct extensive experiments on Poisson's equations, and our comprehensive evaluations have confirmed all of the aforementioned theoretical and numerical advantages. Finally, a mathematically-sound proof affords our new method to generalize to other types of linear PDEs.

## 1 INTRODUCTION

**Background and Major Challenges**. PDEs are quintessential to a wide range of computational problems in science, engineering, and relevant applications in simulation, modeling, and scientific computing. Numerical solutions play an irreplaceable role in common practice because in rare cases do PDEs have analytic solutions, and many general-purpose numerical methods have been made available. Iterative solvers (Saad (2003)) are one of the most-frequently-used methods to obtain a numerical solution of a PDE. Combining iterative solvers with the multigrid method (Briggs & McCormick (2000)) significantly enhances the performance for large-scale problems. Meanwhile, recent deep neural methods have achieved impressive results (Marwah et al. (2021)), yet many currently available neural methods treat deep networks as black boxes. Other neural methods are typically trained in a fully supervised manner on loss functions that directly compare the prediction and the ground truth solution, confining the solution space and resulting in numerical oscillations in the relative residual error even after convergence. These methods generally lack a mathematical backbone, thus offering no guarantee of convergence and correctness. In addition, they may suffer from low accuracy and weak generalization power.

**Motivation and Method Overview**. Partially inspired by the prior work on the structure of multigrid V-cycles (Briggs & McCormick (2000)) and U-Net (Ronneberger et al. (2015)), and to achieve high efficiency and strong robustness, we aim to fully realize neural methods' modeling and computational potential by implanting the legacy numerical methods' mathematical backbone into neural methods in this paper. In order to make our new framework fully explainable, we propose the UGrid framework (illustrated in Fig. 1) based on the structure of multigrid V-cycles for learning the functionality of multigrid solvers. We devise the convolutional operators to incorporate arbitrary boundary conditions

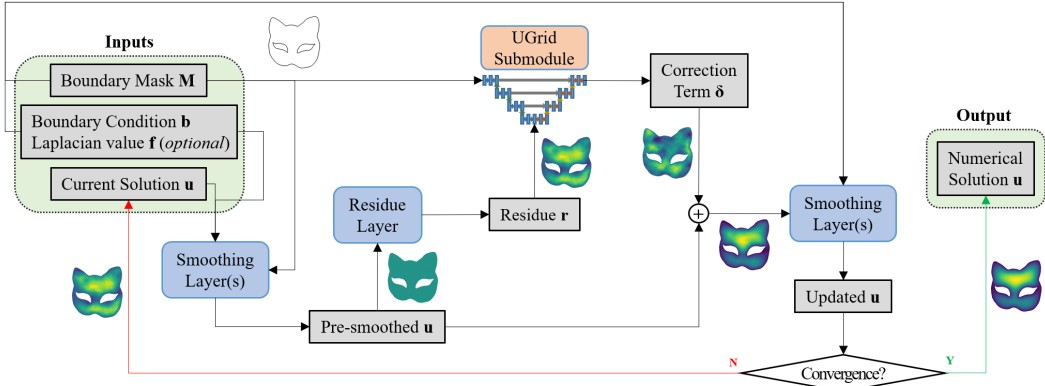

Figure 1: Overview of our novel method. Given PDE parameters and its current numerical estimation, the smoothing operations are applied multiple times first. Then, the current residue is fed into our UGrid submodule (together with the boundary mask). Next, the regressed correction term is applied and post-smoothed several times. Collectively, it comprises one iteration of the neural solver. The UGrid submodule (detailed in Fig. 2) aims to mimic the multigrid V-cycle, and its parameters are learnable, so as to endow our framework with the ability to learn from data.

without modifying the *overall structure* of the key iteration process, and transform the iterative update rules and the multigrid V-cycles into a concrete CNN structure.

**Key Contributions**. The salient contributions of this paper comprise: (1) *Theoretical insight*. We introduce a novel explainable neural PDE solver founded on a solid mathematical background, affording high efficiency, high accuracy, strong generalization power, and a mathematical guarantee to generalize to linear PDEs; (2) *New loss metric*. We propose a residual error metric as the loss function, which optimizes the residue of the prediction. Our newly-proposed error metric enables unsupervised learning and facilitates the unrestricted exploration of the solution space. Meanwhile, it eliminates the numerical oscillation on the relative residual error upon convergence, which has been frequently observed on the legacy mean-relative-error-based loss metrics; and (3) *Extensive experiments*. We demonstrate our method's capability to numerically solve PDEs by learning multigrid operators of Poisson's equations subject to arbitrary boundary conditions of complex geometries and topology, whose patterns are unseen during the training phase. Extensive experiments and comprehensive evaluations have verified all of the aforementioned advantages, and confirmed that our proposed method outperforms the state-of-the-art.

## 2  RELATED WORK

**Black-box-like Neural PDE Solvers**. Much research effort has been devoted to numerically solve PDEs with neural networks and deep learning techniques. However, most of the previous work treats neural networks as black boxes and thus come with no mathematical proof of convergence and correctness. As early as the 1990s, Wang & Mendel (1990a;b; 1991) applied simple neural networks to solve linear equations. Later, more effective neural-network-based methods like Polycarpou & Ioannou (1991); Cichocki & Unbehauen (1992); Lagaris et al. (1998) were proposed to solve the Poisson equations. On the other hand, Wu et al. (1994); Xia et al. (1999); Takala et al. (2003); Liao et al. (2010); Li et al. (2017) used Recurrent Neural Networks (RNNs) in solving systems of linear matrix equations. Most recently, the potential of CNNs and Generative Adversarial Networks (GANs) on solving PDEs was further explored by Tompson et al. (2017); Tang et al. (2017); Farimani et al. (2017); Sharma et al. (2018); Özbay et al. (2021). Utilities used for neural PDE solvers also include backward stochastic differential equations (Han et al. (2018)) and PN junctions (Zhang et al. (2019)).

**Physics-informed Neural PDE Solvers**. Physics-informed Neural Networks (PINNs) have also gained much popularity in recent years. Physics properties, including pressure, velocity (Yang et al. (2016)) and non-locality (Pang et al. (2020)) were used to articulate neural solvers. Mathematical proofs on the minimax optimal bounds (Lu et al. (2022)) and structural improvements (Lu et al.

(2021a;b)) were also made on the PINN architecture itself, endowing physics-informed neural PDE solvers with higher efficiency and interpretability in physics.

**Neural PDE Solvers with Mathematical Backbones**. In 2009, Zhou et al. (2009) proposed a neural-network-based linear system and its solving algorithm with a convergence guarantee. Later on, researchers showed great interest in the multigrid method. Hsieh et al. (2019) modified the Jacobi iterative solver by predicting an additional correction term with a multigrid-inspired linear operator. Greenfeld et al. (2019) proposed to learn a mapping from a family of PDEs to the optimal prolongation operator used in the multigrid Method, which is then extended to Algebraic Multigrids (AMGs) on non-square meshes via Graph Neural Networks (GNNs) by Luz et al. (2020). On the other hand, Li et al. (2021) proposed a Fourier neural operator that learns mappings between function spaces by parameterizing the integral kernel directly in Fourier space. In theory, Marwah et al. (2021) proved that when a PDE's coefficients are representable by small neural networks, the number of parameters needed to better approximate its solution will increase in a polynomial fashion with the input dimension.

## 3 MATHEMATICAL PRELIMINARY

For mathematical completeness, we provide readers with a brief introduction to the concepts that are frequently seen in this paper.

**Discretization of $2$D Linear PDEs**. A linear PDE with Dirichlet boundary condition could be discretized with finite differencing techniques (Saad (2003)), and could be expressed in the following form:

$$\begin{cases} \mathcal{D}u(x,y) = f(x,y), & (x,y) \in \mathcal{I} \\ u(x,y) = b(x,y), & (x,y) \in \mathcal{B} \end{cases}, \tag{1}$$

where $\mathcal{D}$ is a 2D discrete linear differential operator, $\mathcal{S}$ is the set of all points on the discrete grid, $\mathcal{B}$ is the set of boundary points in the PDE, $\mathcal{I} = \mathcal{S} \setminus \mathcal{B}$ is the set of interior points in the PDE, $\partial \mathcal{S} \subseteq \mathcal{B}$ is the set of *trivial boundary points* of the grid.

Using $\mathcal{D}$'s corresponding finite difference stencil, Eq. 1 can be formulated into a sparse linear system of size $n^2 \times n^2$:

$$\begin{cases} (\mathbf{I} - \mathbf{M})\mathbf{A}\mathbf{u} = (\mathbf{I} - \mathbf{M})\mathbf{f} \\ \mathbf{M}\mathbf{u} = \mathbf{M}\mathbf{b} \end{cases}, \tag{2}$$

where $\mathbf{A} \in \mathbb{R}^{n^2 \times n^2}$ is the 2D discrete differential operator, $\mathbf{u} \in \mathbb{R}^{n^2}$ encodes the function values of the interior points and the non-trivial boundary points; $\mathbf{f} \in \mathbb{R}^{n^2}$ encodes the corresponding partial derivatives of the interior points; $\mathbf{b} \in \mathbb{R}^{n^2}$ encodes the non-trivial boundary values; $\mathbf{I}$ denotes the $n^2 \times n^2$ identity matrix; $\mathbf{M} \in \{0,1\}^{n^2 \times n^2}$ is a diagonal binary boundary mask defined as

$$\mathbf{M}_{k,k} = \begin{cases} 1, & \text{Grid point } (i,j) \in \mathcal{B} \setminus \partial \mathcal{S} \\ 0, & \text{Grid point } (i,j) \in \mathcal{I} \end{cases}, \quad k = in + j, \quad 0 \le i, j < n. \tag{3}$$

On the contrary of Eq. 1, both equations in Eq. 2 hold for *all* grid points.

**Error Metric And Ground-truth Solution**. When using numerical solvers, researchers typically substitute the boundary mask $\mathbf{M}$ into the discrete differential matrix $\mathbf{A}$ and the partial derivative vector $\mathbf{f}$, and re-formulate Eq. 2 into the following generic sparse linear system:

$$\widetilde{\mathbf{A}}\,\mathbf{u} = \widetilde{\mathbf{f}}. \tag{4}$$

The *residue* of a numerical solution $\mathbf{u}$ is defined as

$$\mathbf{r}(\mathbf{u}) = \widetilde{\mathbf{f}} - \widetilde{\mathbf{A}}\,\mathbf{u}. \tag{5}$$

In the ideal case, the *absolute residual error* of an exact solution $\mathbf{u}^*$ should be $r_{\mathbf{u}^*} = \|\mathbf{r}(\mathbf{u}^*)\| = 0$. However, in practice, a numerical solution $\mathbf{u}$ could only be an approximation of the exact solution $\mathbf{u}^*$. The precision of $\mathbf{u}$ is evaluated by its *relative residual error*, which is defined as

$$\varepsilon_{\mathbf{u}} = \left\| \widetilde{\mathbf{f}} - \widetilde{\mathbf{A}}\,\mathbf{u} \right\| \Big/ \left\| \widetilde{\mathbf{f}} \right\|. \tag{6}$$

Typically, the ultimate goal of a numerical PDE solver is to seek the optimization of the relative residual error. If we have $\varepsilon_{\mathbf{u}} \le \varepsilon_{\max}$ for some small $\varepsilon_{\max}$, we would consider $\mathbf{u}$ to be a *ground-truth solution*.

**Linear Iterator**. A linear iterator (also called an iterative solver or a smoother) for generic linear systems like Eq. 4 could be expressed as

$$\mathbf{u}_{k+1} = \left(\mathbf{I} - \widetilde{\mathbf{P}}^{-1}\widetilde{\mathbf{A}}\right)\mathbf{u}_k + \widetilde{\mathbf{P}}^{-1}\widetilde{\mathbf{f}}, \tag{7}$$

where $\widetilde{\mathbf{P}}^{-1}$ is an easily invertible approximation to the system matrix $\widetilde{\mathbf{A}}$.

## 4 NOVEL APPROACH

In spite of high efficiency, generalization power remains a major challenge for neural methods. Many SOTA neural solvers, e.g., Hsieh et al. (2019), fail to generalize to new scenarios unobserved during the training phase. Such new scenarios include: (i) New problem sizes; (ii) New, complex boundary conditions and right-hand sides, which includes geometries, topology, and values (noisy inputs); and (iii) Other types of PDEs. Our newly proposed UGrid neural solver resolves problems (i) and (ii), and we provide a mathematical derivation of how UGrid could generalize to other types of linear PDEs. UGrid is comprised of the following two components: (1) The *fixed* neural smoother, which includes our proposed convolutional operators; (2) The *learnable* neural multigrid, which consists of our UGrid module.

### 4.1 CONVOLUTIONAL OPERATORS

**Therotical Insights**. For a specific discrete differential operator $\mathcal{D}$, its corresponding system matrix $\mathbf{A}$ could be expressed as a convolution kernel with known (*fixed*) parameters. In practice, however, $\mathbf{A}$ encodes the boundary geometry and turns into matrix $\widetilde{\mathbf{A}}$ in Eq. 4, and is thus *impossible* to be expressed as a fixed convolution kernel. The introduction of Eq. 2 and the *masked convolutional iterator* seamlessly resolves this problem. Furthermore, we provide a proof of correctness upon convergence (typically circumscribed by neural solvers) for such masked convolutional iterators, and use them as the mathematical backbone of our UGrid solver.

**Masked Convolutional Iterator**. Eq. 7, which is tuned for Eq. 4, could be modified into a *masked version* tuned for Eq. 2. The modification could be *roughly* expressed as "multiplying the right-hand-side of Eq. 7 by $\mathbf{I} - \mathbf{M}$ and adding the product with $\mathbf{M}\mathbf{b}$":

$$\mathbf{u}_{k+1} = (\mathbf{I} - \mathbf{M})\left(\left(\mathbf{I} - \mathbf{P}^{-1}\mathbf{A}\right)\mathbf{u}_k + \mathbf{P}^{-1}\mathbf{f}\right) + \mathbf{M}\mathbf{b}, \tag{8}$$

where $\mathbf{P}$ is an easily-invertible approximation on the discrete differential operator $\mathbf{A}$. (The correctness of Eq. 8 is detailed later.) For 2D Poisson problems, $\mathbf{A}$ could be assembled by the five-point finite difference stencil for 2D Laplace operators (Saad (2003)), and we could simply let $\mathbf{P} = -4\mathbf{I}$, where $\mathbf{I}$ denotes the identity matrix. The update rule specified in Eq. 8 thus becomes

$$\mathbf{u}_{k+1}(i,j) = \frac{1}{4}(\mathbf{I} - \mathbf{M})\left(\mathbf{u}_k(i-1,j) + \mathbf{u}_k(i+1,j) + \mathbf{u}_k(i,j-1) + \mathbf{u}_k(i,j+1) - \mathbf{f}\right) + \mathbf{M}\mathbf{b}. \tag{9}$$

To transform the masked iterator into a convolution layer, we reorganize the column vectors $\mathbf{u}$, $\mathbf{b}$, $\mathbf{M}$ and $\mathbf{f}$ into $n \times n$ matrices with their semantic meanings untouched. Then, the convolution layer could be expressed as

$$\mathbf{u}_{k+1} = \text{smooth}(\mathbf{u}_k) = (\mathbf{1} - \mathbf{M})\left(\mathbf{u}_k * \mathbf{J} - 0.25\mathbf{f}\right) + \mathbf{M}\,\mathbf{b}\,, \quad \mathbf{J} = \begin{pmatrix} 0 & 0.25 & 0 \\ 0.25 & 0 & 0.25 \\ 0 & 0.25 & 0 \end{pmatrix}. \tag{10}$$

**Convolutional Residual Operator**. Except for the smoother, the multigrid method also requires the calculation of the residue in each iteration step. In practice, the residue operator Eq. 5 can also be seamlessly implemented as a convolution layer. Because our masked iterator (Eq. 8) guarantees that $\mathbf{u}$ satisfies $\mathbf{M}\mathbf{u} = \mathbf{M}\mathbf{b}$ at any iteration step, the residue operator could be simplified into

$$\mathbf{r}(\mathbf{u}) = (\mathbf{1} - \mathbf{M})\left(\mathbf{f} - \mathbf{u} * \mathbf{L}\right)\,, \quad \mathbf{L} = \begin{pmatrix} 0 & 1 & 0 \\ 1 & -4 & 1 \\ 0 & 1 & 0 \end{pmatrix}. \tag{11}$$

**Proof of Correntness of Eq. 8**. The following Lemmas and Theorems guarantee that upon convergence, Eq. 8 will yield a ground-truth solution:

**Lemma 1.** *For a fixed linear iterator in the form of*

$$\mathbf{u}_{k+1} = \mathbf{G} \cdot \mathbf{u}_k + \mathbf{c} \, , \tag{12}$$

*with a square update matrice* $\mathbf{G}$ *having a spectral radius* $\rho(\mathbf{G}) < 1$, $\mathbf{I} - \mathbf{G}$ *is non-singular, and Eq. 12 converges for any constant* $\mathbf{c}$ *and initial guess* $\mathbf{u}_0$. *Conversely, if Eq. 12 converges for any* $\mathbf{c}$ *and* $\mathbf{u}_0$, *then* $\rho(\mathbf{G}) < 1$.

*Proof.* Proved as Theorem 4.1 in Saad (2003). □

**Lemma 2.** *For all operator norms* $\|\cdot\|_k$, $k = 1, 2, \ldots, \infty$, *the spectral radius of a matrix* $\mathbf{G}$ *satisfies* $\rho(\mathbf{G}) \le \|\mathbf{G}\|_k$.

*Proof.* Proved as Lemma 6.5 in Demmel (1997). □

**Theorem 1.** *Eq. 8 converges to the ground-truth solution of Eq. 2 when* $\mathbf{P}$ *is full-rank diagonal.*

*Proof.* To prove this Theorem, we only need to prove: (1) Eq. 8 converges to a fixed point $\mathbf{u}$; and (2) The fixed point $\mathbf{u}$ satisfies Eq. 2.

To prove (1), we only need to prove that for the update matrix

$$\mathbf{G} = (\mathbf{I} - \mathbf{M})(\mathbf{I} - \mathbf{P}^{-1}\mathbf{A}) \, ,$$

its spectral radius $\rho(\mathbf{G}) < 1$. As shown in Demmel (1997), the Jacobi iterator converges for a huge variety of linear PDEs with corresponding choices of full-rank diagonal $\mathbf{P}$s. E.g., for 2D Poisson problems, we have $\rho(\mathbf{I} - \mathbf{P}^{-1}\mathbf{A}) < 1$ for $\mathbf{P} = -4\mathbf{I}$. From Lemma 2, taking the *spectral norm* $\|\cdot\|_2$ (i.e., $k = 2$), we have

$$\rho(\mathbf{G}) \le \left\|(\mathbf{I} - \mathbf{M})(\mathbf{I} - \mathbf{P}^{-1}\mathbf{A})\right\|_2 \le \|\mathbf{I} - \mathbf{M}\|_2 \left\|\mathbf{I} - \mathbf{P}^{-1}\mathbf{A}\right\|_2 \, .$$

Furthermore, because $\mathbf{I} - \mathbf{P}^{-1}\mathbf{A}$ is symmetric, we have $\rho(\mathbf{I} - \mathbf{P}^{-1}\mathbf{A}) = \left\|\mathbf{I} - \mathbf{P}^{-1}\mathbf{A}\right\|_2$. On the other hand, because $\mathbf{I} - \mathbf{M} \in \{0, 1\}^{n^2 \times n^2}$ is a binary diagonal matrix, we have $\|\mathbf{I} - \mathbf{M}\|_2 = 1$. This yields $\rho(\mathbf{G}) < 1$.

To prove (2), we first notice that the fixed point $\mathbf{u} = \mathbf{u}_{k+1} = \mathbf{u}_k$ of Eq. 8 satisfies

$$\mathbf{u} = (\mathbf{I} - \mathbf{M})\left((\mathbf{I} - \mathbf{P}^{-1}\mathbf{A})\mathbf{u} + \mathbf{P}^{-1}\mathbf{f}\right) + \mathbf{M}\mathbf{b} \, , \quad \text{i.e.,}$$

$$(\mathbf{I} - \mathbf{M})\mathbf{u} + \mathbf{M}\mathbf{u} = (\mathbf{I} - \mathbf{M})\left((\mathbf{I} - \mathbf{P}^{-1}\mathbf{A})\mathbf{u} + \mathbf{P}^{-1}\mathbf{f}\right) + \mathbf{M}\mathbf{b} \, .$$

Again, since $\mathbf{M} \in \{0, 1\}^{n^2 \times n^2}$ is a binary diagonal matrix, we have

$$\begin{cases} (\mathbf{I} - \mathbf{M})\mathbf{u} = (\mathbf{I} - \mathbf{M})\left((\mathbf{I} - \mathbf{P}^{-1}\mathbf{A})\mathbf{u} + \mathbf{P}^{-1}\mathbf{f}\right) \\ \mathbf{M}\mathbf{u} = \mathbf{M}\mathbf{b} \end{cases} \, . \tag{13}$$

The second equation in Eq. 13 is essentially the second equation in Eq. 2. Furthermore, the first equation in Eq. 13 could be simplified into $(\mathbf{I} - \mathbf{M})\mathbf{P}^{-1}(\mathbf{A}\mathbf{u} - \mathbf{f}) = \mathbf{0}$. Since $\mathbf{P}$ is full-rank diagonal, $\mathbf{P}^{-1}$ should also be full-rank diagonal. Then we have $(\mathbf{I} - \mathbf{M})(\mathbf{A}\mathbf{u} - \mathbf{f}) = \mathbf{0}$, which means that $\mathbf{u}$ also satisfies the first equation in Eq. 2. □

**Generalization to Other Types of Linear PDEs**. In this paper, we choose the 2D Poisson problem as an example for mathematical derivations and qualitative experiments. However, our newly proposed architecture is not for 2D Poisson's equations only. For instance, for steady-state diffusion equations in the form of $\boldsymbol{\nabla} \cdot (k(x, y)\boldsymbol{\nabla}u(x, y)) = f(x, y)$, where $k(x, y)$ denotes a 2D diffusion coefficient field (Poisson's equations are special cases with $k \equiv 1$), we could easily extend Eq. 10 into:

$$\mathbf{u}_{k+1} = (1 - \mathbf{M})\left((\mathbf{u}_k * \mathbf{J}_1 + \left(((\mathbf{k} - \mathbf{k}^i) \circ \mathbf{u}) * \mathbf{J}_2\right) \div 16\mathbf{k}) - 0.25\mathbf{f} \div \mathbf{k}\right) + \mathbf{M}\,\mathbf{b} \, , \tag{14}$$

where $\mathbf{J}_1$ is the regular convolution kernel as in Eq. 10; $\mathbf{J}_2$ is a new convolution kernel $\mathbf{J}_2 = \begin{pmatrix} 0 & 1 & 0 \\ 1 & 0 & -1 \\ 0 & -1 & 0 \end{pmatrix}$; $\mathbf{k}^i$ is a *reordered* version of the coefficient field $\mathbf{k}$, which could be pre-computed; and $\circ$ and $\div$ separately denote *element-wise* matrix multiplication/division. Then, we could apply the same mask operations as in Eq. 10 to get the masked convolution iterator for steady-state diffusion equations. If needed, a further extension to the 3D versions of these problems could also be done with ease.

## 4.2 NEURAL NETWORK DESIGN

**UGrid Iteration**. We design the iteration step of our neural iterator as a sequence of operations as follows (which is illustrated in Fig. 1):

$$
\begin{array}{ll}
\mathbf{u} = \text{smooth}^{\nu_1}(\mathbf{u}) & \text{(Apply pre-smoother for } \nu_1 \text{ times);} \\
\mathbf{r} = \mathbf{r}(\mathbf{u}) & \text{(Calculate the current residue);} \\
\delta = \text{UGrid}(\mathbf{r}, \mathbf{1} - \mathbf{M}) & \text{(Invoke our } \textit{UGrid submodule} \text{ recursively);} \\
\mathbf{u} = \mathbf{u} + \delta & \text{(Apply the correction term);} \\
\mathbf{u} = \text{smooth}^{\nu_2}(\mathbf{u}) & \text{(Apply post-smoother for } \nu_2 \text{ times).}
\end{array}
\tag{15}
$$

The entire iteration process is specifically designed to emulate the multigrid iteration (Saad (2003)): We use the pre-smoothing and post-smoothing layers (as specified in Eq. 10) to eliminate the high-frequency modes in the residue $\mathbf{r}$, and invoke the *UGrid submodule* to eliminate the low-frequency modes.

**UGrid Submodule**. Our UGrid submodule is also implemented as a fully-convolutional network, whose structure is highlighted in Fig. 2. The overall structure of UGrid is built upon the principled combination of U-Net (Ronneberger et al. (2015)) and multigrid V-cycle, and could be considered a "V-cycle" with skip connections. Just like the multigrid method, our UGrid submodule is also invoked recursively, where each level of recursion would coarsen the mesh grid by 2x.

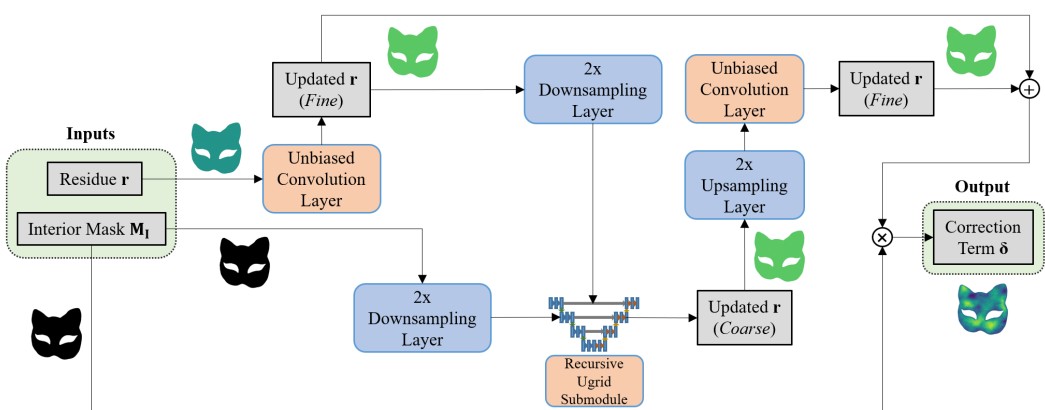

Figure 2: Overview of our recursive UGrid submodule. The residue is smoothed by *unbiased* convolution layers, downsampled to be updated by a 2x-coarser UGrid submodule, then upsampled back to the fine grid, smoothed, and added with the initial residue by skip-connection. Boundary values are enforced by interior mask via element-wise multiplication. The convolution layers (shown in orange) are learnable; other layers (shown in blue) are the fixed mathematical backbone.

To approximate the linearity of the multigrid iteration, we implement the pre-smoothing layers and the post-smoothing layers in the legacy multigrid V-cycle hierarchy (not to be confused with the pre-smoother and the post-smoother in Eq. 15, which are *outside* of the V-cycle hierarchy) as learnable 2D convolution layers *without* any bias. For the same reason, we also drop many commonly-seen neural layers which would introduce non-linearity, such as normalization layers and activation layers.

## 4.3 LOSS FUNCTION DESIGN

**Legacy Loss Metric**. We refer the equivalents of the mean relative error between the predicted value and the ground-truth value as the *legacy loss metric*. Though intuitive, the legacy loss is unstable:

**Theorem 2.** *When a neural network converges on a legacy loss metric such that its prediction* $\mathbf{x}$ *satisfies* $\mathcal{L}_{\text{legacy}}(\mathbf{x}, \mathbf{y}) = \text{mean}\left(|\mathbf{x} - \mathbf{y}|/|\mathbf{y}|\right) \le l_{\max}$, *where* $\mathbf{y}$ *denotes the ground truth value,* $\mathbf{x}$'s *relative residual error still oscillates between* 0 *and an input-dependent maximum value.*

*Proof.* Denote $\varepsilon_{\mathbf{x}}$ as $\mathbf{x}$'s relative residual error, then we have:

$$\varepsilon_{\mathbf{x}} = \frac{\left\|\widetilde{\mathbf{f}} - \widetilde{\mathbf{A}}\,\mathbf{x}\right\|}{\left\|\widetilde{\mathbf{f}}\right\|} \approx \frac{\left\|\widetilde{\mathbf{f}} - \widetilde{\mathbf{A}}\,(\mathbf{y} \pm l_{\max}\,\mathbf{y})\right\|}{\left\|\widetilde{\mathbf{f}}\right\|} = \frac{\left\|\left(\widetilde{\mathbf{f}} - \widetilde{\mathbf{A}}\,\mathbf{y}\right) \mp \left(l_{\max}\,\widetilde{\mathbf{A}}\,\mathbf{y}\right)\right\|}{\left\|\widetilde{\mathbf{f}}\right\|}$$

$$\leq \frac{\left\|\widetilde{\mathbf{f}} - \widetilde{\mathbf{A}}\,\mathbf{y}\right\|}{\left\|\widetilde{\mathbf{f}}\right\|} + \frac{\left\|l_{\max}\,\widetilde{\mathbf{A}}\,\mathbf{y}\right\|}{\left\|\widetilde{\mathbf{f}}\right\|} = \varepsilon_{\mathbf{y}} + \frac{\left\|l_{\max}\,\widetilde{\mathbf{A}}\,\mathbf{y}\right\|}{\left\|\widetilde{\mathbf{f}}\right\|} \, , \tag{16}$$

where $\varepsilon_{\mathbf{y}}$ denotes the relative residual error of the "ground-truth" value $\mathbf{y}$. $\varepsilon_{\mathbf{y}} \neq 0$ because in most cases, a PDE's ground-truth solution could only be a numerical approximation with errors. The maximum value is input-dependent because $\widetilde{\mathbf{A}}$ and $\widetilde{\mathbf{f}}$ are input-dependent. $\qquad\square$

From Theorem 2, we observe that there is **no** guarantee that optimizing the legacy loss metric between $\mathbf{x}$ and $\mathbf{y}$ would increase the precision in terms of the relative residual error. The cause of the observed numerical oscillation is nothing but setting a non-exact numerical approximation as the "ground truth" value. Moreover, the legacy loss metric also restricts the solution space: A numerical solution with low relative residual error may have a large relative difference from the so-called "ground truth" value, and such valid solutions are *unnecessarily* rejected by the legacy loss metric, which clearly indicates one of its another major shortcomings.

**Proposed Residual Loss Metric**. A prediction $\mathbf{x}$ should be considered as a ground truth solution as long as its relative residual error is below the threshold $\varepsilon_{\max}$. We now propose to optimize the neural solver in an unsupervised manner. To be specific, we optimize

$$\mathcal{L}_{\mathbf{r}_{\mathrm{abs}}}(\mathbf{x}) = \mathbb{E}_{\mathbf{x}}[\|(\mathbf{1} - \mathbf{M})(\mathbf{f} - \mathbf{A}\,\mathbf{x})\|_2] \, , \tag{17}$$

which evaluates the absolute residual error in the interior area of the Poisson problem. The reason why we optimize absolute residues instead of relative residues is that: (1) The absolute residues uniformly reflect the relative residues with normalized data, and (2) Our experiments show that adding the relative residue term does not improve performance. It is worth mentioning that the *boundaries* is preserved naturally by our new method, thus **need not** be optimized. Compared to the legacy loss metric, the proposed residual loss metric is closer to the fundamental definition of the precision of a PDE's solution, and is more robust and stable because it does **not** oscillate like the legacy loss metrics. Moreover, the unsupervised training process makes it much easier to gather data and thus achieve better numerical performance.

## 5 EXPERIMENTS AND EVALUATIONS

**Data Generation and Implementation Details**. Our new neural PDE solver is trained in an unsupervised manner on the newly-proposed residual loss. Before training, we synthesized a dataset with $16000$ $(\mathbf{M}, \mathbf{b}, \mathbf{f})$ pairs. To examine the generalization power of our method, the geometries of boundary conditions in our training data are limited to "Donuts-like" shapes as shown in Fig. 3 (h). Moreover, all training data are restricted to Laplacian equations *only*, i.e., $\mathbf{f} \equiv \mathbf{0}$. Our UGrid model has 6 recursive multigrid submodules. We train our model and perform all experiments on a personal computer with 64GB RAM, AMD Ryzen 9 3950x 16-core processor, and NVIDIA GeForce RTX 2080 Ti GPU. We train our model for 300 epochs with the Adam optimizer. The learning rate is initially set to $0.001$, and decays by $0.1$ for every 50 epochs. We initialize all learnable parameters with PyTorch's default initialization policy.

**Experimental Results**. We compare our model with two state-of-the-art legacy solvers, AMGCL (Demidov (2019)), and NVIDIA AmgX (NVIDIA Developer (2022)), as well as one SOTA neural solver proposed by Hsieh et al. (2019). [1] We apply our model and the baselines to the task of

---

[1]AMGCL is a C++ multigrid library with multiple GPU backends, we are comparing with its CUDA backend, with CG solver at the coarsest level. AmgX is part of NVIDIA's CUDA-X GPU-Accelerated Library, and we adopt its official `AMGX_LEGACY_CG` configuration. Hsieh et al. (2019)'s code is available at `https://github.com/ermongroup/Neural-PDE-Solver`. They did not release a pre-trained model, so we train their model with configurations as-is in their training and data-generation scripts, with minimal changes to make the program run.

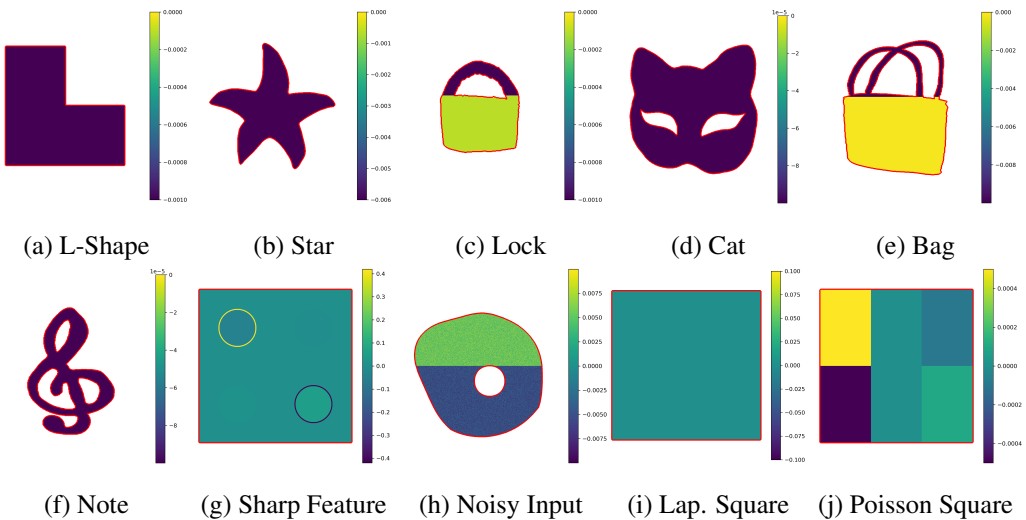

| (a) L-Shape | (b) Star | (c) Lock | (d) Cat | (e) Bag |

| (f) Note | (g) Sharp Feature | (h) Noisy Input | (i) Lap. Square | (j) Poisson Square |

Figure 3: Illustration of the Laplacian distributions of our testcases. The boundaries are shown in bold red lines for a better view. (Boundary values are available in supplemental materials.)

2.5D freeform surface modeling. These surfaces are modeled by Poisson's equations as 2D height fields. Each subfigure in Fig. 3 illustrates the Laplacian distribution of a surface with either trivial or complex geometry/topology. The Poisson equation for each surface is discretized into: (1) Large-scale problem: A linear system of size $1,050,625 \times 1,050,625$, whose qualitative results are documented in Table 1; and (2) Small-scale problem: A linear system of size $66,049 \times 66,049$, whose qualitative results are documented in Table 2.

Our testcases as shown in Fig. 3. These are all with complex geometry and topology, and **none** of which are present in the training data, except the geometry of Fig. 3 (h). Testcase(s) (a-f) examines the strong generation power and robustness of UGrid for irregular boundaries with complex geometries and topology **unobserved** during the training phase; (g) is designed to showcase UGrid's power to handle both sharp and smooth features in one scene (note that there are two sharp-feature circles on the top-left and bottom-right corners, as well as two smooth-feature circles on the opposite corners); (h) examines UGrid's robustness against noisy input (boundary values, boundary geometries/topology, and Laplacian distribution); (i-j) are two baseline surfaces separately modeled by Laplace's equation and Poisson's equation.

Table 1: Comparison of our model and state-of-the-art on large-scale problems. "Time" denotes the time (ms) to reach relative residual errors $\leq 10^{-4}$; "Error" denotes the final relative residual errors, divided by $10^{-5}$. Convergence maps are also available in the supplemental materials.

| Testcase Large-scale | UGrid Time / Error | AMGCL Time / Error | AmgX Time / Error | Hsieh et al. Time / Error |
|---|---|---|---|---|
| Bag | **18.66** / 2.66 | 199.23 / 4.80 | 94.68 / 4.25 | 58.09 / 420 |
| Cat | **10.09** / 2.70 | 276.96 / 6.62 | 114.69 / 6.66 | 49.79 / 14.6 |
| Lock | **10.55** / 9.88 | 140.18 / 4.05 | 67.54 / 4.96 | 49.92 / 55.78 |
| Noisy Input | **10.16** / 2.64 | 262.06 / 3.52 | 116.91 / 9.19 | 51.07 / 2654 |
| Note | **10.31** / 4.06 | 127.28 / 3.00 | 64.35 / 4.56 | 20.26 / 8.67 |
| Sharp Feature | **20.01** / 3.80 | 422.49 / 4.14 | 176.34 / 3.87 | 51.22 / 24883 |
| L-shape | **15.26** / 8.43 | 221.87 / 6.74 | 110.07 / 4.89 | 50.53 / 96.1 |
| Laplacian Square | **15.10** / 3.88 | 422.13 / 3.63 | 188.98 / 9.76 | 31.43 / 9.03 |
| Poisson Square | **15.07** / 9.37 | 414.85 / 5.24 | 189.49 / 9.93 | 50.57 / 974 |
| Star | **15.18** / 7.50 | 152.83 / 6.65 | 72.90 / 5.29 | 50.45 / 384 |

In Table 1, our UGrid model reaches the desirable precision 10-20x faster than AMGCL, 5-10x faster than NVIDIA AmgX, and 2-3x faster than Hsieh et al. (2019). This shows the efficiency and accuracy

of our method. Moreover, the testcase "Noisy Input" showcases the robustness of our solver (i.e., converging efficiently on noisy inputs). Furthermore, among all the ten testcases, only (the geometry of) the "Noisy Input" case is observed in the training phase of UGrid. This shows that UGrid is able to converge to unseen scenarios whose boundary conditions are of complex geometry (e.g., "Cat", "Star", and "L-shape") and topology (e.g., "Note", "Bag", and "Cat"). On the contrary, Hsieh et al. (2019) **failed** to converge in most of our testcases, which verifies one of their claimed limitations (i.e., **no** guarantee of convergence to unseen cases), and showcases the strong generalization power of our method. In addition, even for those converged cases, our method is still faster than Hsieh et al. (2019).

Table 2: Comparison of our model and state-of-the-art on small-scale problems.

| Testcase Small-scale | UGrid Time / Error | AMGCL Time / Error | AmgX Time / Error | Hsieh et al. Time / Error |
|---|---|---|---|---|
| Bag | **8.76** / 8.05 | 12.14 / 3.00 | 22.34 / 8.20 | 47.69 / 252 |
| Cat | 51.96 / 6.21 | **17.03** / 6.98 | 27.66 / 4.83 | 23.02 / 9.95 |
| Lock | **9.00** / 2.11 | 15.77 / 7.89 | 16.96 / 9.36 | 48.72 / 117.9 |
| Noisy Input | **8.94** / 6.00 | 14.00 / 9.39 | 26.30 / 3.14 | 51.79 / 5576 |
| Note | 8.87 / 2.75 | **8.79** / 9.02 | 16.68 / 7.23 | 36.66 / 8.28 |
| Sharp Feature | **13.31** / 7.52 | 21.47 / 4.15 | 49.59 / 6.85 | 49.31 / 24876 |
| L-shape | 40.60 / 7.09 | **12.36** / 9.97 | 24.08 / 9.35 | 50.06 / 96.44 |
| Laplacian Square | **13.21** / 3.27 | 22.22 / 5.60 | 48.60 / 3.98 | 24.57 / 6.54 |
| Poisson Square | **13.21** / 2.88 | 21.93 / 5.51 | 47.56 / 4.03 | 49.77 / 473 |
| Star | **8.92** / 2.36 | 18.93 / 2.17 | 17.96 / 9.42 | 48.68 / 456 |

In Table 2, even on small-scale problems that hinder our solver with a compact multigrid-like hierarchy from delivering its full power, the UGrid model is still faster than or exhibits comparable efficiency with respect to the three SOTA legacy/neural solvers. Again, this shows the high efficiency as well as the strong generalization power of our new method. The testcases "Cat" and "L-shape" showcase that the generalization power (in terms of problem size) does come with a price of potentially downgraded efficiency. Thus, for the sake of the best efficiency, we still recommend re-training UGrid for problems of different sizes.

**Limitations**. As shown in Section 3, our neural PDE solver is designed for linear PDEs *only*, as non-linear PDEs generally have no linear iterative solver and our analysis on the correctness of Eq. 8 does not hold for non-linear PDEs. Another limitation lies in the fact that there is no strict mathematical guarantee on *how fast* our solver will converge (i.e., there is no practical upper bound that confines the maximum number of iterations), which results in similar difficulties for legacy numerical methods. As a consequence, as observed in Table 2, our solver does **not** necessarily converge faster on small grids.

## 6 CONCLUSION AND FUTURE WORK

This paper has articulated a novel efficient-and-rigorous neural PDE solver which is built upon the U-Net and the Multigrid Method, naturally combining the mathematical backbone of correctness and convergence as well as the knowledge gained from data observations. Extensive experiments validate all the self-claimed advantages of our proposed approach. Our future research efforts will be extending the current framework to non-linear PDEs. The critical algorithmic barrier between our approach and non-linear PDEs is the limited expressiveness of the convolution semantics. We would like to explore more alternatives with stronger expressive power.

**Reproducibility Statement**. We are working to the best of our knowledge to make sure that our work can be easily reproduced. The technical details are available in Section 5, the supplemental materials, as well as the README documents of our source code.

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

# A APPENDIX

This supplemental material is provided to readers in the interest of our paper's theoretical and experimental completeness.

## A.1 A BRIEF INTRODUCTION TO THE MULTIGRID METHOD

**Note: This subsection is amended as NEW material.**

The section briefly introduces the multigrid method. We recommend readers who are not familiar with the multigrid method with texts like Saad (2003) and Stüben (2001).

It has been proved that the linear iterators damp the high-frequency components of the residue (Eq. 5) very rapidly. Linear iterators are thus called smoothers. The other components (low-frequency or smooth modes) of the residue are difficult to damp with regular smoothers. However, many of these modes are mapped naturally into high-frequency modes on a coarser mesh. This leads to the multigrid method (Saad (2003)) as detailed in Algorithm 1, which naturally hints at a possible implementation of a similar hierarchy with CNN modules.

---

**Algorithm 1** Multigrid method on Eq. 4: $H$ denotes coarse grid, $h$ denotes fine grid.

---

**Require:** Input $\widetilde{\mathbf{A}}_h, \mathbf{u}_0, \widetilde{\mathbf{f}}_h$; Hyper-parameters $\mathbf{R}_h^H, \mathbf{P}_H^h, \nu_1, \nu_2, \gamma, \varepsilon_{\max}$;
**Ensure:** $\mathbf{u}_h = \mathrm{MG}(\widetilde{\mathbf{A}}_h, \mathbf{u}_0, \widetilde{\mathbf{f}}_h, \nu_1, \nu_2, \gamma, \varepsilon_{\max})$;
  **repeat**
     Pre-smooth: $\mathbf{u}_h \leftarrow \mathrm{smooth}^{\nu_1}(\widetilde{\mathbf{A}}_h, \mathbf{u}_0, \widetilde{\mathbf{f}}_h)$           ▷ Apply pre-smoother for $\nu_1$ times
     Get residual: $\mathbf{r}_h \leftarrow \widetilde{\mathbf{f}}_h - \widetilde{\mathbf{A}}_h \mathbf{u}_h$
     Coarsen: $\mathbf{r}_H \leftarrow \mathbf{R}_h^H \mathbf{r}_h$           ▷ Fine-to-coarse transfer
     Construct coarse system: $\widetilde{\mathbf{A}}_H = \mathbf{R}_h^H \widetilde{\mathbf{A}}_h \mathbf{P}_H^h$
     **if** coarse enough for direct solvers **then**
        Solve: $\widetilde{\mathbf{A}}_H \delta_H = \mathbf{r}_H$
     **else**
        Recursion: $\delta_H \leftarrow \mathrm{MG}^\gamma(\widetilde{\mathbf{A}}_H, \mathbf{0}, \mathbf{r}_H, \nu_1, \nu_2, \gamma - 1)$ ▷ Apply multigrid method for $\gamma$ times
     **end if**
     Correct: $\mathbf{u}_h \leftarrow \mathbf{u}_h + \mathbf{P}_H^h \delta_H$          ▷ Coarse-to-fine transfer
     Post-smooth: $\mathbf{u}_h \leftarrow \mathrm{smooth}^{\nu_2}(\widetilde{\mathbf{A}}_h, \mathbf{u}_h, \widetilde{\mathbf{f}}_h)$     ▷ Apply post-smoother for $\nu_2$ times
  **until** $\|\mathbf{r}_h\| = \left\|\widetilde{\mathbf{f}}_h - \widetilde{\mathbf{A}}_h \mathbf{u}_h\right\| \leq \varepsilon_{\max} \left\|\widetilde{\mathbf{f}}_h\right\|$
  Return $\mathbf{u}_h$

---

One of the keys to the multigrid method is the grid transfer operators. It has been proved that applying the on-the-shelf versions of these operators, e.g., half-weighting restriction operator or linear interpolation (prolongation) operator, is sub-optimal (Greenfeld et al. (2019)). Each specific problem would require one specific operator for best performance. We thus learn correction terms to the legacy grid transfer operators as linear convolution layers.

## A.2 MORE SPECIFICATIONS ON OUR TRAINING DATA

UGrid is trained **only** with pairs of boundary masks and boundary values as shown in Fig. 4 (h). To be specific, throughout the whole training phase, UGrid is exposed only to *zero* Laplacians and piecewise-constant Dirichlet boundary conditions with the "Donut-like" geometries. UGrid is **unaware** of all other complex geometries, topology, as well as the irregular/noisy distribution of boundary values/Laplacians observed in our testcases. This showcases the strong generalization power of our UGrid neural solver.

## A.3 MORE SPECIFICATIONS ON OUR TESTCASES

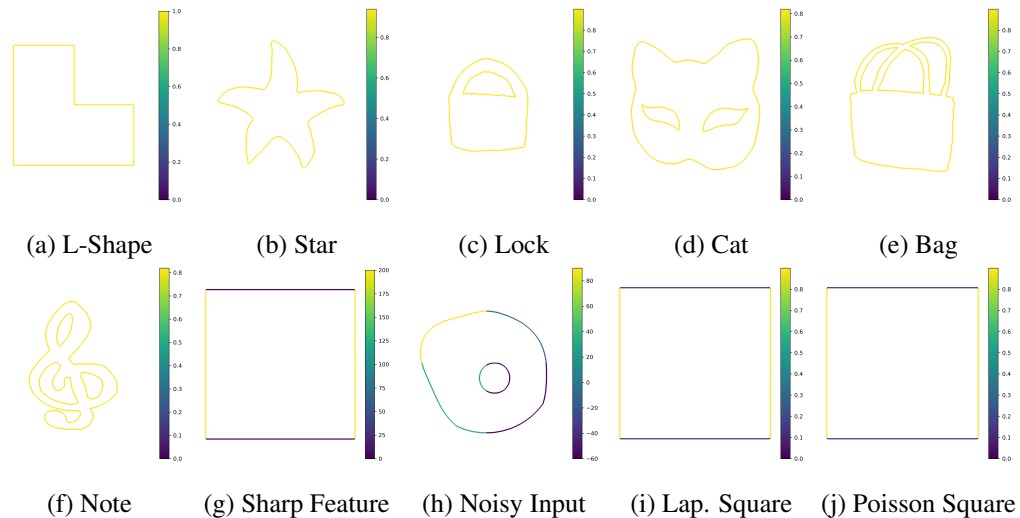

| (a) L-Shape | (b) Star | (c) Lock | (d) Cat | (e) Bag |
| --- | --- | --- | --- | --- |
| (f) Note | (g) Sharp Feature | (h) Noisy Input | (i) Lap. Square | (j) Poisson Square |

Figure 4: Illustration of the Dirichlet boundary values of our ten testcases. Again, all boundaries are shown in bold for a better view. Note that these boundaries are **not** required to be constant and could have *discontinuities*, which could be observed in testcases (g-j).

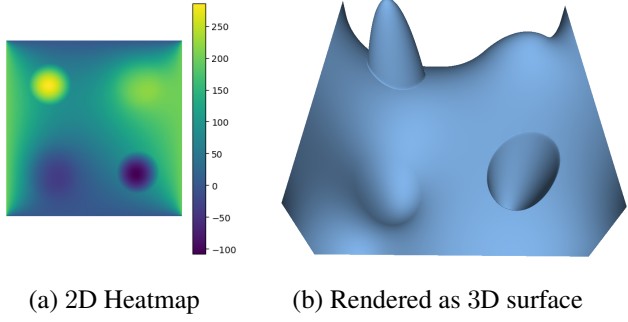

(a) 2D Heatmap      (b) Rendered as 3D surface

Figure 5: The 2.5D height field modeled by testcase "Sharp feature" (Fig. 3 (g) and Fig. 4 (g)). (a) The 2D ground-truth heatmap; (b) The ground truth is rendered as a 3D surface for a better view. Note the sharp bumps at the top-left and bottom-right corners as well as the soft counterparts at the opposite corners. This example illustrates UGrid's capability of preserving both sharp and smooth features in one scene.

## A.4 MORE SPECIFICATIONS ON OUR QUALITATIVE EVALUATIONS

All of our testcases are tested for 100 times and the results are averaged. For UGrid and Hsieh et al. (2019), we set the maximum number of iterations as 64, and the iteration is terminated immediately upon reaching this threshold, no matter whether the current numerical solution has reached the desirable precision. AmgX has no direct support for relative residual errors, so we set tolerance on absolute residual errors case-by-case to achieve similar precision.

For the experimental completeness of this paper, we also provide readers with the convergence maps of UGrid and the three SOTA solvers we compare with. The convergence maps are plotted for all of our ten testcases, each for its two different scales.

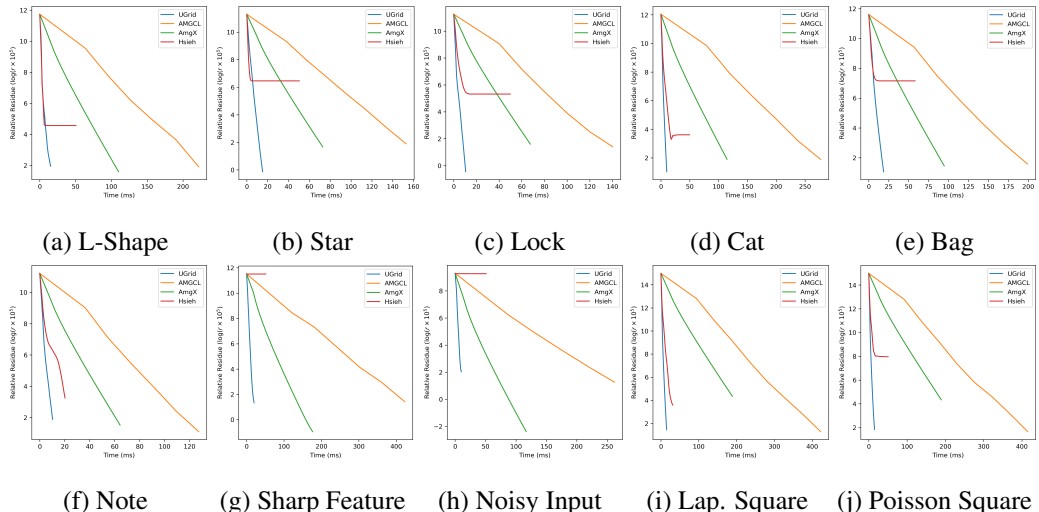

Figure 6: Convergence map on large-scale testcases. The $x$ coordinates are time(s), shown in $\mathrm{ms}$; the $y$ coordinates are the relative residual errors, shown in logarithm $(\log(r \times 10^5))$ for a better view.

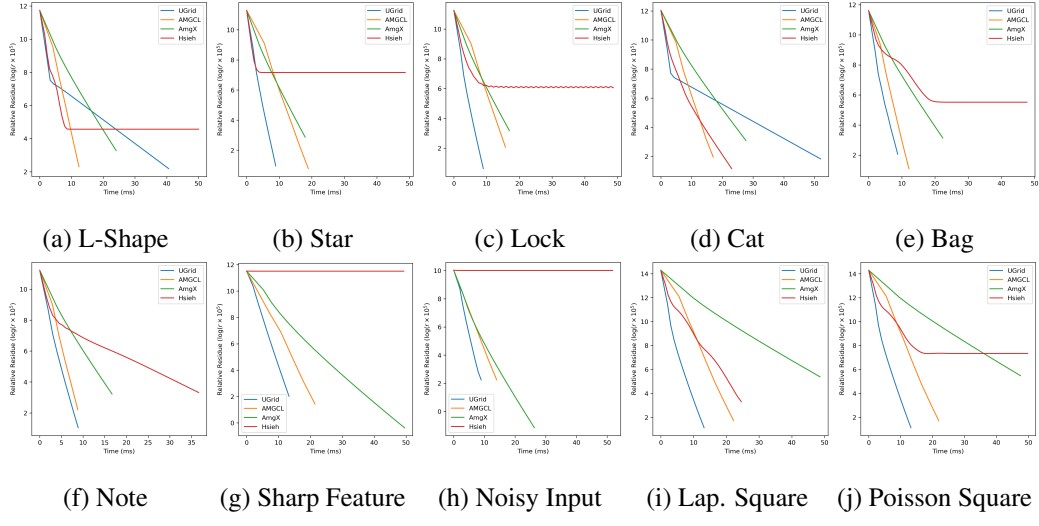

Figure 7: Convergence map on small-scale testcases.

## A.5 ABLATION STUDY

**Note: This subsection is amended as NEW material.**

For experimental completeness, we conduct ablation studies with respect to our residual loss and the UGrid architecture itself. In addition to the UGrid model trained with residual loss (as proposed in the main content), we also train another UGrid model with legacy loss, as well as one vanilla U-Net model with residual loss. This U-Net model has the same number of layers as UGrid, and has non-linear layers as proposed in Ronneberger et al. (2015), We let the U-Net directly regress the solutions to Poisson's equations. All these models are trained with the same data in the same manner (except for the loss metric), as detailed in Section 5.

Reviewers may note that the U-Net model is not trained with the legacy loss. This is because our training data are all generated from Laplacian equations (which, the ability to train on Laplacian and generalize to Poisson, is one of UGrid's technical merits). At present, we don't have enough Poisson data with ground truth values to train such a model on the legacy loss. On the contrary, for

the residual loss, we could simply input randomly-sampled noises as the Laplacian field and optimize the U-Net model with respect to the residue in an unsupervised manner.

We conduct qualitative experiments on the same set of testcases as detailed in Section 5, and the results are as follows:

Table 3: Comparison of UGrid with residual loss, UGrid with legacy loss (UGrid (L)), and U-Net with residual loss, on large-scale problems. "Time" denotes the time (ms) to reach relative residual errors $\leq 10^{-4}$ or for $64$ iterations, whichever comes first; "Error" denotes the final relative residual errors, divided by $10^{-5}$.

| Testcase Large-scale | UGrid Time / Error | UGrid (L) Time / Error | U-Net Time / Error |
|---|---|---|---|
| Bag | **18.66** / 2.66 | 28.81 / 4.86 | 81.71 / 1384131 |
| Cat | **10.09** / 2.70 | 23.80 / 1.43 | 70.09 / 2539002 |
| Lock | **10.55** / 9.88 | Diverge | 70.92 / 1040837 |
| Noisy Input | **10.16** / 2.64 | 20.65 / 2.42 | 73.05 / 21677 |
| Note | **10.31** / 4.06 | Diverge | 69.97 / 614779 |
| Sharp Feature | **20.01** / 3.80 | 31.34 / 5.14 | 70.08 / 222020 |
| L-shape | **15.26** / 8.43 | Diverge | 74.67 / 1800815 |
| Laplacian Square | **15.10** / 3.88 | 30.72 / 2.76 | 72.24 / 30793035 |
| Poisson Square | **15.07** / 9.37 | 31.52 / 3.33 | 71.74 / 31043896 |
| Star | **15.18** / 7.50 | Diverge | 70.01 / 1138821 |

In Table 3, the residual loss endows our UGrid model with as much as 2x speed up versus the legacy loss. The residual loss also endows UGrid to converge to the failure cases of its counterpart trained on legacy loss. These results demonstrate the claimed merits of the residual loss. On the other hand, it will **diverge** if we naively apply the vanilla U-Net architecture directly to Poisson's equations. For experimental completeness only, we list the diverged results in the last column. (The "time" column measures the time taken for $64$ iterations; the iterators are shut down once they reach this threshold.) This showcases the significance of UGrid's mathematically-rigorous network architecture.

Table 4: Comparison of UGrid with residual loss, UGrid with legacy loss (UGrid (L)), and U-Net with residual loss, on small-scale problems. "Time" denotes the time (ms) to reach relative residual errors $\leq 10^{-4}$ or for $64$ iterations, whichever comes first; "Error" denotes the final relative residual errors, divided by $10^{-5}$.

| Testcase Small-scale | UGrid Time / Error | UGrid (L) Time / Error | U-Net Time / Error |
|---|---|---|---|
| Bag | **8.76** / 8.05 | 17.89 / 4.50 | 71.86 / 678141 |
| Cat | **51.96** / 6.21 | Diverge | 68.89 / 1317465 |
| Lock | **9.00** / 2.11 | 18.32 / 2.83 | 69.47 / 189412 |
| Noisy Input | **8.94** / 6.00 | 17.88 / 6.58 | 69.54 / 21666 |
| Note | **8.87** / 2.75 | 17.79 / 3.06 | 69.59 / 24715 |
| Sharp Feature | **13.31** / 7.52 | 26.64 / 1.91 | 70.57 / 191499 |
| L-shape | **40.60** / 7.09 | Diverge | 69.71 / 1011364 |
| Laplacian Square | **13.21** / 3.27 | 22.23 / 9.55 | 73.80 / 15793109 |
| Poisson Square | **13.21** / 2.88 | 22.13 / 9.76 | 71.56 / 15393069 |
| Star | **8.92** / 2.36 | 17.60 / 5.69 | 73.72 / 502993 |

In Table 4, for small-scale problems, the residual loss still endows UGrid with as much as 2x speedup and stronger generalization power against its counterpart trained with legacy loss. Once again, the vanilla U-Net model diverged for all testcases, and we list its diverged results for experimental completeness only.

### A.6 QUALITATIVE EVALUATIONS ON INHOMOGENEOUS HELMHOLTZ EQUATIONS WITH SPATIALLY-VARYING WAVENUMBERS

**Note: This subsection is amended as NEW material.**

Under Dirichlet boundary condition, an inhomogeneous Helmholtz equation with spatially-varying wavenumber may be expressed as follows:

$$\begin{cases} \nabla^2 u(x,y) + k^2(x,y)u(x,y) = f(x,y), & (x,y) \text{ is an interior point} \\ u(x,y) = b(x,y), & (x,y) \text{ is a boundary point} \end{cases}, \tag{18}$$

where $u$ is the unknown scalar field, $k^2$ is the spatially-varying wavenumber, $f$ is the (non-zero) right hand side, and $b$ is the Dirichlet boundary condition. Such an equation is generally **non-self-adjoint**.

For our proposed UGrid solver, we could naturally extend Eq. 10 into the following form to incorporate Eq. 18:

$$\mathbf{u}_{k+1} = \frac{1}{4 - \mathbf{k^2}}(1 - \mathbf{M})(\mathbf{u}_k * 4\mathbf{J} - \mathbf{f}) + \mathbf{Mb} \quad (4 - \mathbf{k^2} \neq 0), \tag{19}$$

where all notations retain their meanings as in Eq. 10; and further extend Eq. 11 into

$$\mathbf{r}(\mathbf{u}) = (1 - \mathbf{M})\big(\mathbf{f} - \mathbf{u} * \mathbf{L} - \mathbf{k^2}\mathbf{u}\big), \tag{20}$$

where all notations retain their meanings as in Eq. 11. We train UGrid with the same training data and residual loss as mentioned in Section 5. As one exception, we also input randomly-sampled $k^2$ during training, evaluation, and testing. The randomly-sampled $k^2$s used for training are illustrated in Fig. 8 (h).

For qualitative experiments, we use the same boundary conditions and Laplacian distributions as shown in Fig. 4 and Fig. 3, and we randomly initialize the wavenumber field $k^2$ across the whole domain, resulting in a noisy distribution:

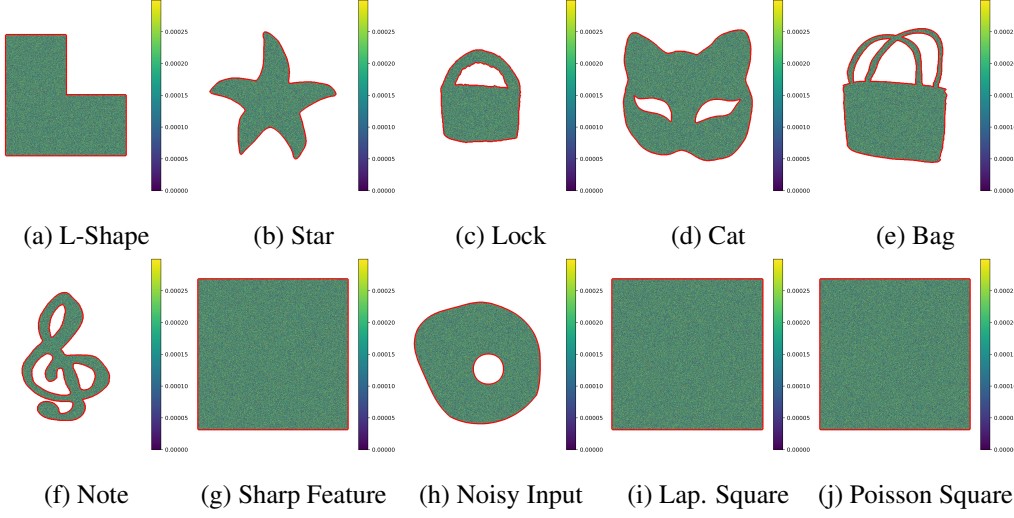

(a) L-Shape     (b) Star     (c) Lock     (d) Cat     (e) Bag

(f) Note     (g) Sharp Feature     (h) Noisy Input     (i) Lap. Square     (j) Poisson Square

Figure 8: Illustration of the wavenumber distributions of our testcases. The boundaries are shown in bold red lines for a better view. (Boundary values are shown in Fig. 4.)

The qualitative results are as follows:

Table 5: Comparison of our model and state-of-the-art on large-scale problems. "Time" denotes the time (ms) to reach relative residual errors $\leq 10^{-4}$; "Error" denotes the final relative residual errors, divided by $10^{-5}$.

| Testcase Large-scale | UGrid Time / Error | AMGCL Time / Error | AmgX Time / Error |
|---|---|---|---|
| Bag | **20.03** / 8.08 | 203.71 / 5.69 | 94.00 / 6.24 |
| Cat | **16.85** / 0.51 | 278.07 / 7.26 | 119.16 / 3.26 |
| Lock | **12.83** / 6.82 | 144.10 / 4.24 | 70.71 / 5.16 |
| Noisy Input | **11.98** / 6.79 | 260.48 / 3.92 | 116.76 / 9.27 |
| Note | **12.18** / 6.48 | 131.58 / 3.02 | 94.47 / 4.73 |
| Sharp Feature | **57.68** / 9.86 | 422.55 / 9.85 | Diverge |
| L-shape | **23.14** / 4.68 | 231.89 / 9.98 | 113.06 / 5.19 |
| Laplacian Square | **44.91** / 9.98 | 419.92 / 5.81 | 188.47 / 7.64 |
| Poisson Square | **43.55** / 8.31 | 421.44 / 9.84 | 193.35 / 8.96 |
| Star | **15.18** / 7.50 | 154.81 / 7.63 | 72.24 / 5.31 |

In Table 5, our UGrid model reaches the desirable precision 7-20x faster than AMGCL and 5-10x faster than NVIDIA AmgX.

Table 6: Comparison of our model and state-of-the-art on small-scale problems. "Time" denotes the time (ms) to reach relative residual errors $\leq 10^{-4}$; "Error" denotes the final relative residual errors, divided by $10^{-5}$.

| Testcase Small-scale | UGrid Time / Error | AMGCL Time / Error | AmgX Time / Error |
|---|---|---|---|
| Bag | 14.70 / 6.29 | **12.92** / 3.00 | 25.82 / 1.79 |
| Cat | **16.63** / 7.86 | 16.82 / 6.98 | 28.37 / 3.03 |
| Lock | **9.78** / 5.87 | 16.23 / 7.88 | 19.75 / 2.02 |
| Noisy Input | 14.95 / 0.76 | **14.34** / 9.40 | 28.92 / 0.04 |
| Note | 14.37 / 8.28 | **9.01** / 9.02 | 18.76 / 2.55 |
| Sharp Feature | **19.46** / 1.18 | 21.37 / 4.21 | 52.82 / 0.13 |
| L-shape | 14.64 / 0.88 | **12.29** / 9.99 | 26.90 / 2.10 |
| Laplacian Square | **14.60** / 4.60 | 22.43 / 5.59 | 43.68 / 17.8 |
| Poisson Square | **15.27** / 6.53 | 22.35 / 5.50 | 43.57 / 17.2 |
| Star | **9.77** / 5.96 | 19.09 / 2.16 | 20.89 / 2.18 |

In Table 6, again, even on small-scale problems that hinder our solver with a compact multigrid-like hierarchy from delivering its full power, UGrid is still faster than or exhibits comparable efficiency with respect to the SOTA.

### A.7 QUALITATIVE EVALUATIONS ON INHOMOGENEOUS STEADY-STATE CONVECTION-DIFFUSION-REACTION EQUATIONS

**Note: This subsection is amended as NEW material.**

Under Dirichlet boundary condition, an inhomogeneous steady-state convection-diffusion-reaction equation may be expressed as follows:

$$\begin{cases} \mathbf{v}(x,y) \cdot \boldsymbol{\nabla} u(x,y) - \alpha \nabla^2 u(x,y) + \beta u(x,y) = f(x,y), & (x,y) \text{ is an interior point} \\ u(x,y) = b(x,y), & (x,y) \text{ is a boundary point} \end{cases}, \quad (21)$$

where $u$ is the unknown scalar field, $\mathbf{v} = (v_x, v_y)^\top$ is the vector velocity field, $\alpha$, $\beta$ are constants, $f$ is the (non-zero) right-hand side, and $b$ is the Dirichlet boundary condition. Such an equation is also generally **non-self-adjoint**.

For our proposed UGrid solver, we could naturally extend Eq. 10 into the following form to incorporate Eq. 21:

$$\mathbf{u}_{k+1} = \frac{1}{4\alpha + \beta}(\mathbf{1} - \mathbf{M})(\alpha\mathbf{u}_k * 4\mathbf{J} + \mathbf{v_x}(\mathbf{u}_k * \mathbf{J_x}) + \mathbf{v_y}(\mathbf{u}_k * \mathbf{J_y}) + \mathbf{f}) + \mathbf{Mb} \quad (4\alpha + \beta \neq 0),$$

(22)

where $\mathbf{J_x} = \begin{pmatrix} 0 & 0 & 0 \\ 0.5 & 0 & -0.5 \\ 0 & 0 & 0 \end{pmatrix}$ and $\mathbf{J_y} = \begin{pmatrix} 0 & -0.5 & 0 \\ 0 & 0 & 0 \\ 0 & 0.5 & 0 \end{pmatrix}$ are two convolution kernels introduced for the gradient operator in Eq. 21, and all notations retain their meanings as in Eq. 10.

Furthermore, we also extend Eq. 11 into

$$\mathbf{r}(\mathbf{u}) = (\mathbf{1} - \mathbf{M})(\mathbf{f} + \mathbf{v_x}(\mathbf{u} * \mathbf{J_x}) + \mathbf{v_y}(\mathbf{u} * \mathbf{J_y}) + \alpha\mathbf{u} * \mathbf{L} - \beta\mathbf{u}),$$

(23)

where all notations retain their meanings as in Eq. 11.

We then train UGrid in the same manner as for Helmholtz equations. As one exception, we input randomly-sampled $\mathbf{v}$s, $\alpha$s, and $\beta$s during training, evaluation, and testing. These values are sampled using the same routine as for Helmholtz equations, resulting in noisy velocity fields like Fig. 8 as well as randomized $\alpha$, $\beta$ coefficients $(4\alpha + \beta \neq 0)$. The qualitative results are as follows:

Table 7: Comparison of our model and state-of-the-art on large-scale problems. "Time" denotes the time (ms) to reach relative residual errors $\leq 10^{-4}$; "Error" denotes the final relative residual errors, divided by $10^{-5}$.

| Testcase Large-scale | UGrid Time / Error | AMGCL Time / Error | AmgX Time / Error |
|---|---|---|---|
| Bag | **41.89** / 4.77 | 198.19 / 5.28 | 106.84 / 1.09 |
| Cat | **100.68** / 9.06 | 270.55 / 9.21 | 138.44 / 1.22 |
| Lock | **58.79** / 4.78 | 135.86 / 4.72 | 70.23 / 3.97 |
| Noisy Input | **84.29** / 8.75 | 256.90 / 4.40 | 122.29 / 0.08 |
| Note | **25.24** / 7.42 | 124.04 / 6.72 | 69.40 / 4.54 |
| Sharp Feature | **33.80** / 7.90 | 407.64 / 4.25 | 191.97 / 0.46 |
| L-shape | **30.09** / 4.70 | 214.45 / 6.28 | 110.21 / 4.53 |
| Laplacian Square | **60.31** / 6.62 | 409.15 / 4.56 | 198.03 / 7.50 |
| Poisson Square | **48.60** / 7.89 | 409.05 / 5.11 | 212.48 / 3.15 |
| Star | **25.59** / 9.38 | 147.47 / 6.01 | 71.75 / 4.16 |

In Table 7, our UGrid model reaches the desirable precision 2-12x faster than AMGCL and on average 2-6x faster than NVIDIA AmgX.

Table 8: Comparison of our model and state-of-the-art on large-scale problems. "Time" denotes the time (ms) to reach relative residual errors $\leq 10^{-4}$; "Error" denotes the final relative residual errors, divided by $10^{-5}$.

| Testcase Small-scale | UGrid Time / Error | AMGCL Time / Error | AmgX Time / Error |
|---|---|---|---|
| Bag | 16.99 / 3.79 | **12.34** / 9.94 | 22.58 / 3.66 |
| Cat | 69.12 / 8.76 | **16.57** / 9.79 | 27.44 / 6.74 |
| Lock | 17.07 / 1.43 | **16.07** / 6.34 | 18.34 / 4.06 |
| Noisy Input | 22.84 / 6.47 | **15.94** / 2.74 | 24.35 / 0.42 |
| Note | 22.76 / 1.17 | **9.38** / 2.67 | 19.25 / 3.79 |
| Sharp Feature | **17.05** / 5.15 | 21.41 / 4.29 | 41.67 / 0.72 |
| L-shape | 35.18 / 6.25 | **13.02** / 2.30 | 25.51 / 3.97 |
| Laplacian Square | 90.73 / 64.5 | **22.14** / 8.17 | 50.49 / 3.50 |
| Poisson Square | 50.95 / 5.01 | **22.05** / 7.34 | 50.06 / 3.28 |
| Star | **17.06** / 3.69 | 18.70 / 7.55 | 18.88 / 4.71 |

In Table 8, again, even on small-scale problems that hinder our solver with a compact multigrid-like hierarchy from delivering its full power, UGrid is still exhibits comparable efficiency with respect to the SOTA. This demonstrates UGrid's generalization power over problem sizes, though possibly at the price of relatively lower efficiency compared to the size it is trained on. Thus, the authors recommend users of UGrid to train it on the actual size of the problems to be solved.

