# OpenReview forum: "Learning An Efficient-And-Rigorous Neural Multigrid Solver"
_ICLR.cc/2024/Conference — Submitted to ICLR 2024_

### Official Review · Reviewer_BHtd · 2023-10-31

**Soundness:** 3 good
**Presentation:** 2 fair
**Contribution:** 3 good
**Rating:** 5
**Confidence:** 4

**Summary:**

This paper introduces a data-driven multigrid solver for Poisson equations and presents a comparative analysis with the state-of-the-art algebraic multigrid solver, AMGCL, implemented in CUDA. However, the reviewer has identified several conceptual errors related to multigrid methods and the discretization of partial differential equations in the presentation. As a result, it is challenging to provide a recommendation for acceptance at this time.

**Strengths:**

- The method proposed utilizes the natural architectural resemblance of an MG V-cycle and the U-Net (thought this is more like a strength of Hsieh et al. ICLR 2019 paper).
- How to deal with irregular boundaries with unstructured mesh is an important problem in AMG.

**Weaknesses:**

- On page 3, the authors wrote "the ultimate goal of a numerical PDE solver is to seek the optimization of the relative residual", then on page 6, the authors went on to implement the smoother as parametrized layers to get optimized. There is a huge missing link here. The "optimal" smoother should have a desired "smoothing property" that damps the "high frequency" components on a grid efficiently. While directly throwing everything into a loss makes the optimal smoother more likely to be damping all the frequencies. No studies, either theoretical or experimental, have made toward this.
- Page 5, "Jacobi iterator converges for a huge variety of linear PDEs", there is plainly wrong. If Jacobi is sufficient, why use Gauss Seidel, or more advanced block smoothers ( block Schwarz preconditioner to be precise) for discretization for Maxwell's equations?
- Page 6 says "invoke UGrid **recursively**": notation-wisely, there is no recursion defined in this algorithm at all.
- Page 6, Theorem 2, what the authors proved is not "the relative residual ***oscillates***..." at all. If one ought to prove "oscillatory" behavior, the inequality should be actually reversed, for example, use $\|r\|_{\infty}$ greater than something after certain iterations, or use $\limsup$ as what is shown in Gibbs phenomenon.
- In all experiment, $u$ is harmonic with right-hand side being zero, and judging from the boundary conditions shown in Figure 4, all the $u$'s are smooth. While it is okay to do this for testing a direct solver, this practice is utterly insufficient for testing the performance of an iterative solver.
- On page 9 The authors wrote "there is no strict mathematical guarantee on how fast our solver will converge, which result in similar difficulties in for legacy numerical methods". This is so blatantly wrong, in fact, with a known smoother such as Jacobi or GS, MG convergence will be taught in a decent numerical PDE class. There are too many references on this, please check for example, [Hackbusch].
- Regarding the comments on non-linear PDE solvers, "as nonlinear PDEs generally have no linear iterative solver", this is again blatantly wrong that multigrid solvers have been developed for nonlinear equation for a long time, for example, see [Xu1994SISC], and also more recent development in AMG such as [Brandt2012BAMG] by Achi Brandt himself. Even the more advanced FAS can be traced back to the Math. Comp. 1977 paper of Brandt [Brandt1977].
- References:

[Brandt1977]: A. Brandt. Multi-Level Adaptive Solutions to Boundary-Value Problems. Math. Comp. 1977.

[Xu1994SISC]: J. Xu. A novel two-grid method for semilinear equations, SIAM J. of Sci. Comp. 1994.

[Brandt2012BAMG]: A. Brandt et al. Bootstrap AMG. SIAM J. of Sci. Comp. 2011.

[Hackbusch]  W. Hackbusch. [Multi-Grid Methods and Applications](https://link.springer.com/book/10.1007/978-3-662-02427-0). Springer, 1985.

**Questions:**

- On page 4, it says "$P^{-1}$ is an easily invertible approximation of $A$", while a few lines below, it says "$P$ is an easily invertible approximation of $A$".
- I cannot understand why the authors make a big deal about the mask matrix $M$, every practitioner in using multigrid to solve equations arising from FEM/FDM eliminates the unknowns associated with boundary degrees of freedom first to apply the solver only on interior unknowns.
- Is there any nonlinearity (activation) at all in the UGrid submodule? If not, what is the point of using a neural network?
- Suppose the coarse grid problem is solved exactly by a two-grid scheme of the proposed method, let us say $I - B^{-1}A := S^{\nu_2} (I - P(P^{\top}AP)^{-1}P^T A ) S^{\nu_1}$, where $S= I - RA$ is the smoother, and $P$ and $P^T$ are the projection and prolongation operator, respectively, please write down more clearly that which part is parametrized.
- In the experiment, no details about how AMGCL is set up is given. If the time of building the MG hierarchy is counted toward the time of AMGCL, then I guess it is better to include the training time in UGrid.
- In Figure (3) g, the authors used the caption "sharp feature", I simply cannot fathom what is "sharp" here. Are the region of the two circles included in the computational domain of $-\Delta u = 0$? Do they have a discontinuous diffusion coeffcient?
- No information on what the mesh is like, this is important even for designing AMG. If the grid is uniform, then one can learn  prolongation and restriction operators in the geometric multigrid. If it is AMG, the coarsening can be trained. See the references below.
- Missing references:
  - Juncai He and Jinchao Xu. Mg-Net: A unified framework of multigrid and convolutional neural network. arXiv:1901.10415
  - Taghibakhshi et al. Optimization-Based Algebraic Multigrid Coarsening Using Reinforcement Learning. NeurIPS 2021.
  - Alexandr Katrutsa, Talgat Daulbaev, Ivan Oseledets. Deep Multigrid: learning prolongation and restriction matrices. [ arXiv:1711.03825](https://arxiv.org/abs/1711.03825)

---

> ### Author Response · Authors · 2023-11-16
> **Responses to Reviewer BHtd (Part 1/2)**
>
> **Q1. The "optimal" smoother should damp "high frequency", while throwing everything into a loss makes the optimal smoother more likely to be damping all frequencies.**
>
> **R1**. (1) Our smoothing operator (Eq. 10) is **not** learnable. It serves as our mathematical backbone. UGrid's learnable part is a correction term to legacy grid transfer operators. Since we do not optimize smoothers, we won't "make the optimal smoother more likely to be damping all the frequencies". (2) UGrid does preserve high-frequency components, which is shown in testcase "Sharp Feature" (Fig. 5).
>
> **Q2. "Jacobi iterator converges for a huge variety of linear PDEs" is wrong.**
>
> **R2**. Sorry for the confusion. It's true that Jacobi does not converge in many situations. That’s why we state "a huge variety", not "most/all". Moreover, this paper handles Poisson & other simple PDEs, and Jacobi does converge to these PDEs. More specifically, under Dirichlet boundary condition, Jacobi generally converges to well-behaved elliptic PDEs & many other linear PDEs like time-variant diffusion/convection equations in homogeneous materials. Thus, this paper is compatible to a variety of PDEs.
>
> **Q3. There is no recursion defined in this algorithm at all.**
>
> **R3**. Please refer to R5 to Reviewer pjtp (above).
>
> **Q4. For "oscillatory behavior", Thm. 2 should prove $|r|_{\infty} >$ something.**
>
> **R4**. (1) Sorry for the misinterpretation. Thm 2 shows the possible max error when optimizing legacy loss, which explains the existance of solutions with small relative residuals but large legacy errors. Please also refer to R1 to Reviewer Ejzh (ablation study on residual/legacy losses). (2) Denote the prediction as $x$, the ground truth as $y$, $\varepsilon$s as their stochastic relative residuals, and $l_{max}$ as the legacy error between $x$ and $y$:
> $$
> \varepsilon_{x} =
> \dfrac{||f - A \ x||}{||f||} \approx
> \dfrac{||f - A \ (y \pm l_{max} y)||}{||f||} =
> \dfrac{||(f - A \ y) \mp l_{max} \ A \ y)||}{||f||} \ge
> \left|\dfrac{||f - A \ y||}{||f||} - \dfrac{||l_{max} \ A \ y||}{||f||}\right| =
> \left|\varepsilon_y - \dfrac{||l_{max} \ A \ y||}{||f||}\right| > 0
> \ .
> $$
>
> **Q5. In experiments, $u$ is harmonic (with RHS being zero) and smooth?**
>
> **R5**. Sorry for the confusion, but we should clarify that $u$ is neither harmonic nor smooth. **(1) Not Harmonic**. $u$ is harmonic only for our training data, and 1/10 of our testcases (Fig. 3 (i)). All other testcases have non-zero Laplacians. **(2) Non-zero RHS**. See Fig. 4. None of the testcases have zero Dirichlet boundary values. Fig. 4 (g-j) have non-uniform boundary values. **(3) Not Smooth**. Fig. 3 (c) (e) (h) (j) have jumps in Laplacians (thus C(1) only). For Poisson’s equations, $u$ must be C(1) in inner part (Laplacian operator involves second-order derivatives). $u$ could be discontinuous on the boundary: Testcase "Sharp Feature" (see Fig. 5) introduces "boundary elements" in the inner part of $u$, making $u$ only C(0) ($u$ is defined both inside & outside the circles, and $u$ is treated as a whole; we employ no boundary/interior segmentation.). Testcase "Noisy Input" (Fig. 4 (h)) has discontinous boundary values, making $u$ not even C(0).
>
> **Q6. "No strict mathematical guarantee on how fast legacy methods will converge" is wrong.**
>
> **R6**. Sorry for the misinterpretation, we meant to say that "there is no uniform upper bound which could be used to estimate the exact time needed without testing". This is because convergence guarantees are typically asymptotic, not always reachable in practice, e.g., conjugate gradient method often converges faster than what its $O(\sqrt{k})$ complexity suggests.
>
> **Q7. It's wrong to say that "nonlinear PDEs generally have no linear iterative solver".**
>
> **R7**. We are sorry for the misinterpretation. We would conduct literature reviews more thoroughly.
>
> **Q8. On page 4, which is the easily-invertible approx. of $A$, $P^{-1}$ or $P$?**
>
> **R8**. $P$ is the easily invertible approx. of $A$, and $P^{-1}$ is the inverse of $P$.
>
> **Q9. Why the authors make a big deal about the mask matrix M?**
>
> **R9**. (1) M enables us to treat all discrete grid points in the same way, without the need of segmentation. This simplifies the problem. (2) Without M, the Jacobi update kernel will change w.r.t. input geometry, adding more burden to the network. We did not have M in the earliest version of our work, and the network diverged.

---

> ### Author Response · Authors · 2023-11-16
> **Responses to Reviewer BHtd (Part 2/2)**
>
> **Q10. Is there any nonlinearity (activation) at all in the UGrid submodule? If not, what is the point of using a neural network?**
>
> **R10**. Please refer to R3 (for not using activation) and R10 (for using convolution) to Reviewer pjtp (above).
>
> **Q11. Which parts in the multigrid are parametrized (take a two-grid scheme as an example)?**
>
> **R11**. (1) UGrid does not involve a direct solver at the coarsest level; it applies the smoothers directly. Thus, it's not precise to map UGrid to a two-grid cycle with an exact solver. (2) UGrid's learnable convolution operators are correction terms to grid transfer operators. Thus, we would state that $P (P^{\top} A P)^{-1} P^{\top}$ is parametrized as a whole.
>
> **Q12. If time for building MG hierarchy for AMGCL/AmgX is included, then training time for UGrid should also be included?**
>
> **R12**. (1) We do include MG hierarchy building time. Yet, it's not fair to include the training time for UGrid, because: (1.1) Training is required only once for one type of PDEs, and could be ignored over the solver's lifespan; (1.2) AMGCL/AmgX must reconstruct their MG hierarchy when input grid or boundary geometry changes; UGrid doesn't need retraining for these cases; (2) When MG hierarchy is constructed already, and only RHS changes, AMGCL/AmgX performs better. However, in fields like PDE-based CAD, grid and boundary-geometry changes as frequently as RHS, and UGrid will be a better choice.
>
> **Q13. "Sharp feature" in Fig. 3 (g) does not seem sharp; are there discontinuous effects? Are the region of the two circles included in the computational domain? Do they have a discontinuous diffusion coefficient?**
>
> **R13**. Please refer to R5 (above) and Fig. 5 (on pp.13).
>
> **Q14. No information on what the mesh is like. If the grid is uniform, then one can learn prolongation and restriction operators in the geometric multigrid. If it is AMG, the coarsening can be trained.**
>
> **R14**. We use sub-uniform unstructured mesh. The grid points are distributed uniformly inside an irregular region with non-trivial geometry and topology, so the grids are unstructured. It is far from trivial to apply GMG directly on our unstructured testcases. Existing GMG solvers either deal with structured boundaries or employ complicated grid transfer operators which are hard to implement on GPU (e.g. quad-tree-based operators). To the best of our knowledge, there is no GPU-based GMG solver which could handle our testcases, and UGrid outperforms existing AMG solvers.

---

> > ### Comment · Reviewer_BHtd · 2023-11-22
> >
> > The authors have taken steps to address some of my concerns, leading me to increase my score. However, it's worth noting that a well-implemented Geometric Multigrid (GMG) or Algebraic Multigrid (AMG) approach may still outperform the proposed method.

---

> > > ### Author Response · Authors · 2023-11-23
> > > **Official Response to Reviewer BHtd**
> > >
> > > Thank you for your feedback! Though a legacy MG solver fine-tuned for spefcific PDE formulations might achieve similar or higher efficiency than UGrid, there is no such solver publicly available at present, and we achieve much higher efficiency compared to SOTA general-purpose solvers. Furthermore, fine tuning such legacy solvers is a non-trivial task requiring a strong mathematical background and non-trivial programming effort. UGrid offers people without such profession a decent solver, only at a one-time cost of data generation (without need of ground truth solution) and training, which is one of the technical merits of unsupervised neural methods. Moreover, just like fine-tuning general-purpose solvers, UGrid could also be further optimized in the following aspects: (1) for the coarest-level solution, we could replace the current iterative routine with more optimized methods, like conjugate gradient solvers or direct solvers; and (2) we could explore more efficient smoothing routines and replace our current Jacobi-based smoother, etc. UGrid is open to optimizations and could also be made more efficient. We appreciate your feedback and would kindly ask for your favor on this work.

---

### Official Review · Reviewer_pjtp · 2023-10-31

**Soundness:** 4 excellent
**Presentation:** 1 poor
**Contribution:** 2 fair
**Rating:** 3
**Confidence:** 3

**Summary:**

The paper proposes a neural network based approach to solve poisson, for different boundaries, boundary conditions, forcings, based on an interpretation of multigrid. The contribution is

**Strengths:**

I like where this paper is coming from and globally I think that combining NNs and more classical methods interesting and promising direction.

Emphasis has been put on generalising outside of the training distribution, and generalizing to quite a wide range of variables (albeit most of the generalisation comes by construction).

**Weaknesses:**

The paper is not well written; i feel the exposition could be greatly simplified. For example, not all reviewers should have to be familiar with multigrid methods, and this could have been explained in the appendix.

The related work (although quite thorough) section is meant to contrast your work wrt to the literature, explain differences and similarities. E.g. it would be nice to clearly state the differences wrt Hsieh et al 2019.

Neural is mentioned in the title, Unet is mentioned in the abstract, but from my understanding the learning part consists in learning convolutions, as there are no activation functions?

From the best of my understanding, the convergence result does not apply to the multigrid method with learned convolutions. What is the link between eq 8 and your algorithm?

In section 4.2, in the UGrid Iteration, it would be nice to have a formal algorithm describing what is happening. As far as I understand, the Ugrid function is called recursively on the same grid, whereas in Fig. 2 this is not the case? There are two different types of smoothing? All this is very confusing.

Not enough details regarding the experimental section.

How is the data generated? How can we evaluate the difficulty of solving the system and assess the usefulness of the method if this information is missing?

I would have been nice to compare with a standard Unet that regresses directly to the solution, as well as learning the whole process with the *legacy loss metric*

**Questions:**

Are you backpropagating the loss through the iterative method?

What is the motivation behind learning the convolutions? If the smoothing operation removes the high-frequencies, why not simply have a downsampling operation?

---

> ### Author Response · Authors · 2023-11-16
> **Responses to Reviewer pjtp (Part 1/2)**
>
> **Q1. The paper is not well written; I feel the exposition could be greatly simplified. E.g., not all reviewers are familiar with multigrid. Explain in appendix.**
>
> **R1**. We have added more explanations on the multigrid method on pp.12 (Section A.1) of the rebuttal-revision pdf. Moreover, we would like to clarify that numerical approaches for PDEs, including the multigrid method, as well as their integration with AI-based methods, are of fundamental significance to scientific fields. Pointers regarding these topics are available in the main content of the paper, including pp.1 (first two paragraphs of Section 1), which explains the significance of these methods; and pp.6 (Section 4.2), which details our implementation of a multigrid-like structure with CNN.
>
> **Q2. Differences w.r.t. Hsieh et al. (2019).**
>
> **R2**. **(1) Loss**. UGrid is trained with residual loss; Hsieh et al. is trained with legacy loss, which is subject to numerical oscillations (Thm. 2); **(2) Learnable Components**. Hsieh et al. learns a linear operator H, which works on the finest grid only; UGrid has learnable components on all grid levels; **(3) Training Procedure**. Hsieh et al. executes their model for one iteration on a preconditioned intermediate result, and optimizes their update vs legacy MG; while UGrid is applied on a raw input for multiple iterations, and we optimize the final residual error; **(4) MG Depth**. Hsieh et al. only leverage a 3-level MG; UGrid reaches as deep as 6 levels. These differences endow UGrid with larger solution space (generalization power), higher efficiency, and higher precision, which are shown in our experiments.
>
> **Q3. "Neural" and "U-Net" mentioned, yet UGrid learns convolutions with no activation functions.**
>
> **R3**. UGrid has no activation functions. It mimics the linear property of legacy solvers and serves as our mathematical backbone. Moreover, Neural Networks could still learn from data without non-linear layers; we admit such a NN coallapses to one single layer. However, this is exactly what we want: We are learning to invert a system matrix ($y = A^{-1} x$).
>
> **Q4. What’s the link between convergence of UGrid and convergence of Eq. 8?**
>
> **R4**. (1) UGrid's "convergence guarantee" indicates "if the iterator converges, its result is correct", which is not observed in most neural methods. The UGrid iteration takes Eq. 8 as pre- and post-smoothers; the last operation in a UGrid iteration is post-smoothing, and convergence test of UGrid also evaluates the convergence of Eq. 8. Thus, convergence guarantee of Eq. 8 ensures the correctness of UGrid's output. The convergence guarantee makes our algorithm more mathematically-rigious. (2) Convergence itself (strictly on math) is guaranteed for Eq. 8 (Thm 1). For the whole UGrid routine, it's evaluated by our experiments.
>
> **Q5. More explanation on the recursive calls on UGrid submodule. Is the UGrid function called recursively on the same grid? Are there two different types of smoothers?**
>
> **R5**. (1) UGrid is an iterative solver. In its iteration step (Eq. 15), "smooth" refers to Eq. 10, "r" refers to Eq. 11, "UGrid" refers to UGrid submodule (Fig. 2). The UGrid submodule is a CNN and it's better to be visualized than to be expressed in math. (2) UGrid submudule is invoked recursively on a coarser grid, **not** the same grid (see the bottom of Fig. 2, note the preceeding down-sampling layers). (3) There is only one type of smoother (Eq. 10). UGrid's learnable part is not a smoothing operator, it is designed as a correction term to the legacy grid transfer operators (i.e., down- and up-sampling layers).
>
> **Q6. More details regarding the experimental section.**
>
> **Q7**. Please refer to: (1) supplemental materials; (2) R1 & R5 to Reviewer Ejzh (above).
>
> **Q8. How is the data generated?**
>
> **R8**. Our training data are Laplacian equations, thus we only need to generate boundary geometries and values. The geometries are generated from circles (with random distortions to mimic random simple closed curves), and we put an extra "hole" inside each curve. The boundary values are piecewise constants. For more details, please refer to "UGrid/script/generate.sh" in our anomynous repo. We would also like to clarify that our testcases have much more complicated boundary geometries/topology/values, and Laplacian distributions, which is illustrated in Fig. 3 and Fig. 4.
>
> **Q9. Ablation study.**
>
> **R9**. Please refer to R1 to Reviewer Ejzh (above).

---

> ### Author Response · Authors · 2023-11-16
> **Responses to Reviewer pjtp (Part 2/2)**
>
> **Q10. Are you backpropagating the loss through the iterative method?**
>
> **R10**. Yes, and for stability during the training process, we are setting the number of iterations to a fixed number (64) during training.
>
> **Q11. Why learn convolutions?**
>
> **R11**. (1) Many differential operators are analogous to convs, e.g., discrete Laplacian operator is a 3x3 conv kernel. This leads to (2) Many components in legacy iterative solvers are also analogous to convolution and thus could be easily implemented with conv layers.
>
> **Q12. If the smoothing operation removes the high-frequencies, why not simply have a downsampling operation?**
>
> **R12**. The downsampling operator (grid-transfer operator) is a tricky component in the multigrid hierarchy as the optimal operator differs for different PDEs and boundaries. UGrid's learnable parts are correction terms on legacy grid transfer operators.

---

### Official Review · Reviewer_APYK · 2023-10-31

**Soundness:** 3 good
**Presentation:** 3 good
**Contribution:** 3 good
**Rating:** 6
**Confidence:** 3

**Summary:**

This paper presents a mathematically solid neural PDE solver, which combines iterative solvers and the Multigrid Method with Convolutional Neural Networks (CNNs). The proposed UGrid neural solver leverages an integration of of U-Net and MultiGrid, with a mathematically sound proof of both convergence and correctness. The proposed model demonstrates appealing results on both numerical accuracy and generalization capabilities on complex situations not encountered during training. The model is training in a unsupervised style with a novel loss. The experiments on Poisson's equations can verify the advantages claimed in this paper. The proposed method has the potential to generalize to other linear PDEs supported a mathematically-sound proof.

**Strengths:**

- The paper provides a novel explainable neural PDE multigrid solver. The explainability of the model make it has a generalization ability, which can be applied to problems with new sizes/boundaries/RHS without re-training. In addition,  the proposed method has the potential to generalize to other types of linear PDEs (e.g., steady-diffusion) (though it seems that there are no quantitative results to support this claim)
- The proposed masked convolutional iterator is intuitive and supported by a proof of correctness (though I didn't go through the proof in detail, it seems to be sensible to me after looked through )
- The results in the experiment part show that the proposed method can outperform or achieve competitive accuracy comparing to SOTA numerical multigrid solvers on both large and small scale problems with less time (may not true on small scale problems).

**Weaknesses:**

- The proposed method can only be applied to linear PDEs currently. It is unclear how the method can be extend to non-linear PDEs as the core proof of the correctness for masked convolutional iterator is not hold for non-linear cases.
- The model is only validated on 2D Poisson's equations though with variety in sizes, boundaries etc. One key contribution the paper claim is the generalization ability to other types of linear PDEs, however there seems no quantitative experiments to support this claim.
- The experiments results on small scale problems are somehow diverge between different test cases (see questions part below). And some cases requires even more time to coverage comparing to large scale problems, which looks counter-intuitive as the computational cost of traditional numerical solvers usually decrease when the problem sizes go smaller.

**Questions:**

- The proposed model requires a significant long time (2 to 3 times comparing to numerical solvers AMGCL and AmgX) to converge on small-scale 'cat' and 'L-shape' cases. In the paper, the authors claim that it si the price of generalizing to new problems regarding to problem size. I am wondering what are the size differences between the training data (i.e., the Donuts-like case) and the 'cat' and 'L-shape'?
- Why does it only happens on these two specific cases but other cases with complex geometries such as 'Bag', 'Star' etc? Does it because the size of other test cases are more similar to the training data?
- It seems that the proposed model perform better (both in accuracy and efficiency) on large scale to small scale problems comparing to the traditional numerical solvers. Do you have any intuitive explanations about this?
- Just wondering have you tried applied the proposed method on other linear PDEs such as steady-state diffusion equations for generalization ability test?

---

> ### Author Response · Authors · 2023-11-16
> **Responses to Reviewer AYPK**
>
> **Q1. It is unclear how the method can be extended to non-linear PDEs.**
>
> **R1**. Yes, and we leave the extension to non-linear PDEs as future work.
>
> **Q2. No quantitative experiments to support generalization to other types of linear PDEs.**
>
> **R2**. Please refer to R5 to Reviewer Ejzh (above).
>
> **Q3. UGrid takes longer to converge on small 'cat' and 'L-shape'. Why does it only happen in these two cases? Does 'cat'/'L-shape' have more similiar problem sizes to the training data than other small cases?**
>
> **R3**. (1) Not all small-scale cases take longer. UGrid performs well on all other eight small-scale testcases (Fig. 3 (b-c) (e-j)). The small "cat"/"L-shape" are outliers. (2) Problem size: Training data are all large-scaled systems of size 1,050,625x1,050,625; small-scale data are of size 66,049x66,049; and "cat" and "L-shape" do not differ from other small cases. (3) Numerical issues are always tricky. It’s hard to analyze efficiency when the lower-level problem, generalization power to unseen problem sizes, is itself more or less relying upon experience, but far less mathematically strict. We would leave this problem to future work.
>
> **Q4. Why UGrid performs better on large scale to small scale problems compared to the traditional numerical solvers?**
>
> **R4**. **(1) Why better than legacy solvers**: Legacy solvers are general-purpose solvers which do not exploit the special properties of specific PDEs, e.g., legacy multigrid solvers use sub-optimal general-purpose grid transfer operators for all PDEs. UGrid is trained on specific PDEs, and it adjusts the grid transfer operation at each level towards the optimal. **(2) Why better on large scale**: UGrid is trained on the large scale, so it reaches its best potential here. Generalization to small scale is tested by our experiments, not mathematically guaranteed in a strict sense. We leave the analysis of the mathematical details on the small-scale testcases as future work. **(3) Traditional MG solvers** may become faster if their components (e.g. grid transfer operator) are tailored for specific PDE formulations. It is not possible to ask a single conference paper to do a full justice on all of the above-mentioned criteria and their in-depth theoretical analysis and guarantees. We leave explorations on these possibilities as future work.
>
> **Q5. Have you tried applied UGrid on other linear PDEs?**
>
> **R5**. Please refer to R5 to Reviewer Ejzh (above).

---

### Official Review · Reviewer_Ejzh · 2023-11-01

**Soundness:** 3 good
**Presentation:** 3 good
**Contribution:** 1 poor
**Rating:** 3
**Confidence:** 4

**Summary:**

- The paper proposes a novel neural PDE solver called UGrid that integrates convolutional neural networks with the iterative multigrid method.
UGrid is designed to solve linear PDEs and provides mathematical guarantees on convergence and correctness. This sets it apart from many other neural PDE solvers that lack theoretical grounding.
- The architecture consists of a fixed neural smoother module and a learnable UGrid submodule. The smoother eliminates high-frequency error modes and UGrid mimics a multigrid V-cycle to reduce low-frequency errors.
- A new residual loss function is proposed that enables training and explores the solution space more freely compared to typical mean squared error losses.
- Experiments on 2D Poisson equations with complex geometries and topologies unseen during training demonstrate UGrid's efficiency, accuracy, and generalization ability compared to state-of-the-art baselines.
- The mathematical framework provides a pathway to generalize UGrid to other linear PDEs beyond Poisson's equation.
Limitations include restriction to linear PDEs only and no strict upper bound on the convergence rate.

In summary, the key innovation is the integration of mathematical principles into the neural network architecture to create an interpretable and provably correct PDE solver with strong performance. The residual loss and unsupervised training are also notable contributions.

**Strengths:**

1. Novel integration of multigrid methods with deep learning - This is an innovative approach that combines mathematical principles with data-driven modeling. The interplay between multigrid components and neural networks is interesting.
2. Interpretable architecture - The method is more explainable than black-box neural solvers, with modules that mimic identifiable parts of the multigrid pipeline. This interpretability could be further enhanced.
3. Generalization capabilities - The results demonstrate some ability to generalize to unseen geometries and topologies during training. This is promising and hints at the potential for broader generalization, which can be further explored.
4. Unstructured mesh support - By operating directly on matrix representations, the approach can handle unstructured grids and complex geometries. This is more flexible than CNNs on Euclidean grids.
5. Outperforms legacy solvers - The experiments show compute time improvements over non-ML solvers. With tuning, the advantage over-optimized methods could become substantial.

**Weaknesses:**

- There are no ablation studies analyzing the effects of key components like the UGrid submodule and the residual loss function. Are these additions really critical? How much do they each improve performance over a baseline?
- The multigrid interpretation of the UGrid module is hand-wavy. A more rigorous analysis of how it mimics multigrid components like restriction, prolongation and coarse-grid correction would enhance the scientific merit.
- The generalization claims are overstated. Testing on different geometries is not enough to prove generalization over problem sizes, PDE types and discretization schemes. Much more rigorous experimentation is needed.
- The computational efficiency gains seem quite modest in practice. Is the added model complexity worth 2-3x speedups? How does training time factor in?

**Questions:**

See the weaknesses. The only way to change my opinion is if I see results in anything other than Poisson (preferably non-self-adjoint equations).

---

> ### Author Response · Authors · 2023-11-16
> **Responses to Reviewer Ejzh (Part 1/2)**
>
> **Q1. Ablation study on the effects of UGrid & residual loss.**
>
> **R1**. We trained another UGrid with legacy loss (UGrid (L)), and a vanilla U-Net with residual loss, which regresses solution to Poisson's equations directly. The experimental results are as follows:
>
> | Testcase | UGrid | UGrid (L) | U-Net |
> | :---: | :---: | :---: | :---: |
> | **(Large-scale)** | **Time (ms) / Error (1e-5)** | **Time (ms) / Error (1e-5)** | **Time (ms) / Error (1e-5)** |
> | Bag | **18.66** / 2.66 | 28.81 / 4.86 | 81.71 / 1384131 |
> | Cat | **10.09** / 2.70 | 23.80 / 1.43 | 70.09 / 2539002 |
> | Lock | **10.55** / 9.88 | Diverge | 70.92 / 1040837 |
> | Noisy Input | **10.16** / 2.64 | 20.65 / 2.42 | 73.05 / 21677 |
> | Note | **10.31** / 4.06 | Diverge | 69.97 / 614779 |
> | Sharp Feature | **20.01** / 3.80 | 31.34 / 5.14 | 70.08 / 222020 |
> | L-shape | **15.26** / 8.43 | Diverge | 74.67 / 1800815 |
> | Laplacian Square | **15.10** / 3.88 | 30.72 / 2.76 | 72.24 / 30793035 |
> | Poisson Square | **15.07** / 9.37 | 31.52 / 3.33 | 71.74 / 31043896 |
> | Star | **15.18** / 7.50 | Diverge | 70.01 / 1138821 |
>
>
> | Testcase | UGrid | UGrid (L) | U-Net |
> | :---: | :---: | :---: | :---: |
> | **(Small-scale)** | **Time (ms) / Error (1e-5)** | **Time (ms) / Error (1e-5)** | **Time (ms) / Error (1e-5)** |
> | Bag | **8.76** / 8.05 | 17.89 / 4.50 | 71.86 / 678141 |
> | Cat | **51.96** / 6.21 | Diverge | 68.89 / 1317465 |
> | Lock | **9.00** / 2.11 | 18.32 / 2.83 | 69.47 / 189412 |
> | Noisy Input | **8.94** / 6.00 | 17.88 / 6.58 | 69.54 / 21666 |
> | Note | **8.87** / 2.75 | 17.79 / 3.06 | 69.59 / 24715 |
> | Sharp Feature | **13.31** / 7.52 | 26.64 / 1.91 | 70.57 / 191499 |
> | L-shape | **40.60** / 7.09 | Diverge | 69.71 / 1011364 |
> | Laplacian Square | **13.21** / 3.27 | 22.23 / 9.55 | 73.80 / 15793109 |
> | Poisson Square | **13.21** / 2.88 | 22.13 / 9.76 | 71.56 / 15393069 |
> | Star | **8.92** / 2.36 | 17.60 / 5.69 | 73.72 / 502993 |
>
> Residual loss endows UGrid with 2x speed up, and more importantly, stronger generalization power (UGrid (L) diverged on several cases). The U-Net model **failed to converge**, which demonstrates the significance of UGrid's mathematically-rigorous architecture. More details are available in the rebuttal-revision pdf (pp.14~).
>
> **Q2. How UGrid mimics multigrid components?**
>
> **R2**. **(1) Smoother**: UGrid implements it as masked convolutional iterator (Eq. 8); **(2) Grid Transfer Operator**: UGrid mimics it with legacy operators plus correction terms (learnable convolution layers); **(3) Coarsest-level Solution**: UGrid does not invoke direct solvers, as those are not as trivially-differentiable as iterative solvers. We invoke Eq. 8 directly on the coarsest level, and **it works**.
>
> **Q3. Generalization claims overstated. Geometry tests not enough to prove generalization over problem sizes, PDE types and discretization schemes.**
>
> **R3**. (1) We claimed that UGrid generalizes to: (1.1) Boundary geometries/values unseen during training phase (validated by our testcases with different geometries); and (1.2) New problem sizes (validated by the small-scale testcases; note that UGrid is trained at the large scale); and (2) If we have other types of PDEs, we must have UGrid retrained. Please refer to R5 (below).
>
> **Q4. Efficiency gain is modest. Added model complexity not worth 2-3x speedup (over Hsieh et al.). How does training time factor in?**
>
> **R4**. (1) Hsieh et al. **failed to converge** to most of our testcases and its time is the time taken before it reaches max number of iterations. When we compare UGrid with another solver, it must CONVERGE on the testcases. "2-3x speedup" does **not** reflect UGrid's efficiency gain. The overall gain should be 10-20x (vs AMGCL) and 5-10x (vs AmgX). (2) Training time. Training is required only once for one type of PDEs. It should be treated as preprocessing time and could be ignored over the long lifespan of the model.

---

> ### Author Response · Authors · 2023-11-16
> **Responses to Reviewer Ejzh (Part 2/2)**
>
> **Q5. Experiments on other PDEs (preferably non-self-adjoint equations).**
>
> **R5**. We have conducted experiments on two more non-self-adjoint equations: (1) Inhomogeneous Helmholtz equations with spatially-varying coefficients, i.e., $\nabla^2 u + k^2 u = f$, where $k^2$ is a non-zero spatially-varying coefficient, and $f$ is a non-zero right-hand side; (2) Inhomogeneous diffusion-convection-reaction equations, i.e., $\mathbf{v} \cdot \mathbf{\nabla} u - \alpha \nabla^2 u + \beta u = f$, where $\mathbf{v}$ is a varying vector velocity field, $\alpha$, $\beta$ are constants, and $f$ is the non-zero right-hand side.
>
> Results for inhomogeneous Helmholtz equations (with spatially-varying coefficients) are as follows:
>
> | Testcase (Helmholtz) | UGrid | AMGCL | AmgX |
> | :---: | :---: | :---: | :---: |
> | **(Large-scale)** | **Time (ms) / Error (1e-5)** | **Time (ms) / Error (1e-5)** | **Time (ms) / Error (1e-5)** |
> | Bag | **20.03** / 8.08 | 203.71 / 5.69 | 94.00 / 6.24 |
> | Cat | **16.85** / 0.51 | 278.07 / 7.26 | 119.16 / 3.26 |
> | Lock | **12.83** / 6.82 | 144.10 / 4.24 | 70.71 / 5.16 |
> | Noisy Input | **11.98** / 6.79 | 260.48 / 3.92 | 116.76 / 9.27 |
> | Note | **12.18** / 6.48 | 131.58 / 3.02 | 94.47 / 4.73 |
> | Sharp Feature | **57.68** / 9.86 | 422.55 / 9.85 | Diverge |
> | L-shape | **23.14** / 4.68 | 231.89 / 9.98 | 113.06 / 5.19 |
> | Laplacian Square | **44.91** / 9.98 | 419.92 / 5.81 | 188.47 / 7.64 |
> | Poisson Square | **43.55** / 8.31 | 421.44 / 9.84 | 193.35 / 8.96 |
> | Star | **15.18** / 7.50 | 154.81 / 7.63 | 72.24 / 5.31 |
>
>
> | Testcase (Helmholtz) | UGrid | AMGCL | AmgX |
> | :---: | :---: | :---: | :---: |
> | **(Small-scale)** | **Time (ms) / Error (1e-5)** | **Time (ms) / Error (1e-5)** | **Time (ms) / Error (1e-5)** |
> | Bag | 14.70 / 6.29 | **12.92** / 3.00 | 25.82 / 1.79 |
> | Cat | **16.63** / 7.86 | 16.82 / 6.98 | 28.37 / 3.03 |
> | Lock | **9.78** / 5.87 | 16.23 / 7.88 | 19.75 / 2.02 |
> | Noisy Input | 14.95 / 0.76 | **14.34** / 9.40 | 28.92 / 0.04 |
> | Note | 14.37 / 8.28 | **9.01** / 9.02 | 18.76 / 2.55 |
> | Sharp Feature | **19.46** / 1.18 | 21.37 / 4.21 | 52.82 / 0.13 |
> | L-shape | 14.64 / 0.88 | **12.29** / 9.99 | 26.90 / 2.10 |
> | Laplacian Square | **14.60** / 4.60 | 22.43 / 5.59 | 43.68 / 17.8 |
> | Poisson Square | **15.27** / 6.53 | 22.35 / 5.50 | 43.57 / 17.2 |
> | Star | **9.77** / 5.96 | 19.09 / 2.16 | 20.89 / 2.18 |
>
> UGrid exhibits a 7-20x speedup vs AMGCL and 5-10x speedup vs AmgX on large-scale problems, and generalization power to unseen small-scale problem sizes, with comparable efficiency w.r.t. AMGCL/AmgX.
>
> Results for inhomogeneous steady-state diffusion-convection-reaction equations are as follows:
>
> | Testcase (Diffusion) | UGrid | AMGCL | AmgX |
> | :---: | :---: | :---: | :---: |
> | **(Large-scale)** | **Time (ms) / Error (1e-5)** | **Time (ms) / Error (1e-5)** | **Time (ms) / Error (1e-5)** |
> | Bag | **41.89** / 4.77 | 198.19 / 5.28 | 106.84 / 1.09 |
> | Cat | **100.68** / 9.06 | 270.55 / 9.21 | 138.44 / 1.22 |
> | Lock | **58.79** / 4.78 | 135.86 / 4.72 | 70.23 / 3.97 |
> | Noisy Input | **84.29** / 8.75 | 256.90 / 4.40 | 122.29 / 0.08 |
> | Note | **25.24** / 7.42 | 124.04 / 6.72 | 69.40 / 4.54 |
> | Sharp Feature | **33.80** / 7.90 | 407.64 / 4.25 | 191.97 / 0.46 |
> | L-shape | **30.09** / 4.70 | 214.45 / 6.28 | 110.21 / 4.53 |
> | Laplacian Square | **60.31** / 6.62 | 409.15 / 4.56 | 198.03 / 7.50 |
> | Poisson Square | **48.60** / 7.89 | 409.05 / 5.11 | 212.48 / 3.15 |
> | Star | **25.59** / 9.38 | 147.47 / 6.01 | 71.75 / 4.16 |
>
>
> | Testcase (Diffusion) | UGrid | AMGCL | AmgX |
> | :---: | :---: | :---: | :---: |
> | **(Small-scale)** | **Time (ms) / Error (1e-5)** | **Time (ms) / Error (1e-5)** | **Time (ms) / Error (1e-5)** |
> | Bag | 16.99 / 3.79 | **12.34** / 9.94 | 22.58 / 3.66 |
> | Cat | 69.12 / 8.76 | **16.57** / 9.79 | 27.44 / 6.74 |
> | Lock | 17.07 / 1.43 | **16.07** / 6.34 | 18.34 / 4.06 |
> | Noisy Input | 22.84 / 6.47 | **15.94** / 2.74 | 24.35 / 0.42 |
> | Note | 22.76 / 1.17 | **9.38** / 2.67 | 19.25 / 3.79 |
> | Sharp Feature | **17.05** / 5.15 | 21.41 / 4.29 | 41.67 / 0.72 |
> | L-shape | 35.18 / 6.25 | **13.02** / 2.30 | 25.51 / 3.97 |
> | Laplacian Square | 90.73 / 64.5 | **22.14** / 8.17 | 50.49 / 3.50 |
> | Poisson Square | 50.95 / 5.01 | **22.05** / 7.34 | 50.06 / 3.28 |
> | Star | **17.06** / 3.69 | 18.70 / 7.55 | 18.88 / 4.71 |
>
> UGrid exhibits a 2-12x speedup vs AMGCL and on average 2-6x speedup vs AmgX on large-scale problems, and generalization power to unseen small-scale problem sizes, with comparable efficiency w.r.t. AMGCL/AmgX.
>
> These results demonstrate UGrid’s generalization power to elliptic PDEs, and in principle, as long as the Jacobi iterator converges, any linear PDE with a convolution-compatible differential stencil should be readily solvable by UGrid. For more details, please refer to our rebuttal-revision pdf (pp.16~).

---

### Meta-Review · Area_Chair_4cMj · 2023-12-10

**Metareview:**

The paper proposes a neural multigrid solver, which focuses on a 2D Poisson equation discretized on a rectangular grid.
It also describes the architecture and the loss function for the proposed solver, together with the experiments

Strenghts:
1) Reported speedups
2) Motivation to build rigorous solvers, the idea is actually quite reasonable.

Weaknesses:
1) No motivation behind the loss function (17), which is described in a vague way
2) Mix of classical well-known results, used as a theory, new proposal and related work (some of it related to multigrid just missing).
3) The problem is quite limited (2D Poisson), where there exist a very good and old baseline, namely https://www.sciencedirect.com/science/article/pii/037704279090252U) which will beat all neural networks by a large margin

**Justification For Why Not Higher Score:**

The paper is not written well, and still has to be significantly improved.

**Justification For Why Not Lower Score:**

N/A

---

### Decision · Program_Chairs · 2024-01-16

Reject